# Assessment of probability distributions and minimum storage draft-rate analysis in the equatorial region

Hasrul Hazman Hasan[1], Siti Fatin Mohd Razali[1], Nur Shazwani Muhammad[1], Firdaus Mohamad Hamzah[2]

[1]Department of Civil Engineering, Faculty of Engineering & Built Environment, Universiti Kebangsaan Malaysia, Bangi, 43600, Malaysia
[2]Department of Engineering Education, Faculty of Engineering & Built Environment, Universiti Kebangsaan Malaysia, Bangi, 43600, Malaysia

*Correspondence to*: Siti Fatin Mohd Razali (fatinrazali@ukm.edu.my)

**Abstract.** Rapid urbanisation in the state of Selangor, Malaysia have led to a change in the land use, the physical properties of basins, vegetation cover and impermeable surface water. These changes have affected the pattern and processes of the hydrological cycle resulting in the ability of the basin region to store water supply to decline. Reliability on water supply from rivers basin depends on their low flow characteristics. Thus, this study is essential to understand the concept of low flow,

drought characteristics, and the predictive significance of river storage-draft rates in managing sustainable water catchment. In this study, the long-term streamflow data of 40-years from seven stations in Selangor were used, and streamflow trends are analysed. Low flow frequency analysis was derived using the Weibull plotting position and four specific frequency distributions. Maximum likelihood was used to parameterise, while Kolmogorov-Smirnov tests are used to evaluate their fit to the dataset. The mass curve is used to quantify the minimum storage draft-rate required to maintain the 50% mean annual

flow for 10-years recurrence interval of low flow. Next, low flow river discharges were analysed using 7-day mean annual minimum while drought event was determined using the 90th percentile (Q90) as the threshold level. The moving average was employed to remove the dependent and minor droughts in determining the drought characteristics. The result of the study shows that the Log-normal (2P) distribution was found to be the best fit for low flow frequency analysis to derive low flow return period. This analysis reveals that September to December is a critical period in river water storage to sustain the water

availability during low flow in a 10-year occurrence interval. The results indicated the hydrological droughts have generally become more frequent and critical in the availability of rivers to sustain water demand during low flows. These results can help in emphasising the natural flow of water to provide water supply for continuous use during low flow.

## 1 Introduction

Droughts are long-term natural disaster phenomena resulting from less-than-average precipitation causing significant damages

to a wide variety of sectors, affecting large regions. The rapid development of the world now sees an increase in population,

and climate change tends to increase drought occurrences (Bakanoğullari and Yeşilköy, 2014; Tigkas et al., 2012). Droughts have considerable economic, societal, and environmental impacts. Drought can typically be classified into four types, depending on different kinds of impacts of drought in different areas: meteorological, hydrological, agricultural and socio-economic (Hasan et al., 2019; Tri et al., 2019). Any type of drought is dynamic and defined by various characteristics such as frequency, severity, duration, and magnitude. The main factor involved in hydrological drought is climate change and anthropogenic activities of surface water resources. The hydrological drought assessment gives a good interpretation of the water surface of the hydrological cycle. Hydrological drought also allows the incorporation of spatial details that impact internal storage and soil, vegetation and terrain characteristics. This study mainly focuses on hydrological drought. The related hydrological aspects, including low water levels and decreased groundwater recharge, are more directly affected by the hydrological drought impacts.

Extreme drought can cause significant water cycle imbalances that alter the processes of precipitation and evaporation, the circulation of atmospheric water vapour and the availability of soil moisture, which results in a low volume of water in streams, rivers and reservoirs. The equilibrium between both the water that is taken out for supply and that is substituted by surface runoff must be maintained. A critical issue arises when there is a dry season, and there is no estimated water excess. Under such conditions, water shortages can happen even though the dry season is not too extreme. Drought is most frequently the consequence of climate change and human activities in the particular area or regions. Human activities and poor management of water resources are exacerbated and exacerbated by water scarcity and drought. In certain regions, water consumption increases the severity of water scarcity and triggers water shortage events in regions that are relatively well endorsed with water resources (Wada et al., 2013).

Hydrological drought is a natural event with streamflow deficits in duration and volume (Kubiak-Wójcicka and Bąk, 2018). In hydrological drought, not every low flow occurrence can be called a drought, and several low flows can form one hydrological drought (Teegavarapu et al., 2019). It is not advisable to equate hydrological drought with low flow or other related hazards. Low flow is a term that is often used, referring to low flow discharge. Low flow is often defined by minimum annual series which does not reflect hydrological drought in all years. Fleig et al. (2006) distinguished between hydrological drought and low flow characteristics. For some specific purposes, the main feature of drought is said to be the water deficit. Low flows are usually observed during a drought, but they only feature one aspect of the drought, namely the magnitude of drought. Low flow analysis is described as analyses that attempt to understand the short-term physical development of flows at a point along a river. The minimal annual n-day average discharge is the most widely used low flow index.

The hydrological drought design system is somewhat complicated and susceptible to catchment characteristics or climate, and a combination of the two variables (Loon et al., 2015; Mohammed and Scholz, 2018; Zhai and Tao, 2017). Precipitation and temperature are two main factors among different environmental factors that mainly determine the climate model and

antecedent situation for hydrological drought events (Joetzjer et al., 2013). Water availability in many areas is becoming less predictable due to climate change. More significant periods of drought and higher temperature are projected to affect the rainfall distribution, river flow used for water availability causing deleterious effects on water supply. Watershed also performs a significant part in the propagation of drought and affects procedures such as pooling, lagging, and lengthening (Fleig et al., 2006; Sarailidis et al., 2019). Some researches further explored the specific functions of climate control and watershed

influence in regulating features of hydrological drought, and the findings are hugely based on spatial scales (Austin and Nelms, 2017; Barker et al., 2016; Liu et al., 2012; Zarafshani et al., 2016; Zhu et al., 2018). Generally, the duration of hydrological drought and the quantity of the deficit are more climate-related than watershed-related. However, watershed features such as geology, region, slope, and groundwater regime perform a significant part in regulating the duration of hydrological drought and the quantity deficit for regional scale where the climate is presumed to be relatively constant (Gianfagna et al., 2015; Laaha

and Blöschl, 2006, 2007; Liu et al., 2016). The influences on hydrological drought are not restricted to the external variables such as climatic and watershed variables and should not be disregarded for anthropogenic activities in the form of land-use modification, reservoir control, irrigation, and water extraction or withdrawal (Hatzigiannakis et al., 2016; Richter and Thomas, 2007; Sun et al., 2018; Toriman et al., 2013).

Water storage in river basins is typically affected by its composition and physical features, such as the morphology of the basin and channel, and type of geological and topographical conditions of the basin (Costa et al., 2003; Robin Burgers et al., 2014). While the mechanics of depletion processes of water storage are generally well understood, modelling of quantitative storage behaviour patterns is rarely possible due to lack of knowledge of storage properties such as geometry, porosity and hydraulic conductivity, the absence of data on evapotranspiration rates, and the transition between storage and loss of storage. It is fair

to idealise the physical situation in these situations, even at the basin scale. All storages, except the storage of channels, are regarded as separate and independent components of different types. They are assumed to contribute a net inflow to the stream, and it is assumed that streamflow response depends on the time elapsed rather than the actual input time.

In the events that the low flow of the river is sufficient to meet the water demand, the storage may be utilised to increase the

guaranteed water supply. The hydrological aspects which must be considered are the amount of storage necessary to sustain a given draft rate and the associated risk of insufficient storage to meet this draft rate. The relationship between inflow, storage and draw-off is complex. The significant sources of error are associated with frequency analysis. Error in frequency analysis is due to fitting the type of extreme value distribution to low flow series and uncertainties associated with assigning recurrences interval for cumulative probabilities to the events in series. Drainage basin stores are surface of significant quantities of water

that may regulate the rate at which input feeds through to the output. Channel storage is the volume of water contained within banks of the river that will operate as a water store between its initial input and ultimate output (Griffiths and Clausen, 1997).

This study was conducted at Selangor states on the west coast of Peninsular Malaysia to evaluate and investigate the hydrological drought characteristics using historical streamflow data. High demand for water that can accommodate the daily water consumption of the population due to rapid populations, as well as the lack of rain, has caused disruptions of water supply in Selangor (Khalid, 2018; Kwan et al., 2013; Ngang et al., 2017). Water shortages associated with the incident of El Nino / Southern Oscillation (ENSO) impacted parts of Malaysia, including Selangor (Sanusi et al., 2015; Zainal et al., 2017). Drought disasters have hit several regions in Malaysia, especially in the Klang-Selangor Valley, Penang and several other places such as Kedah, Kelantan, Sarawak and Sabah (Chan, 2012). The problems of water shortage and drought in Malaysia have been recorded as early as 1951, where it occurred for 29 months in the Langat River Basin (Chan, 2012). After that episode, the drought disaster continued to hit Malaysia with the Klang Valley water crisis in February - May 1998, the water shortage continued in Hulu Langat Selangor in 2002 (Ithnin, 2014). This drought has caused the water level in some water dams in Peninsular Malaysia to reach critical levels, like what happened in the 1997-1998 drought episode (Lee et al., 2018). Consequently, the characteristics of hydrological drought must be identified, and the effects of hydrological drought quantitatively evaluated. Studies conducted by Iqbal et al. (2016), Azadi et al. (2018), and Tigkas et al. (2012) have highlighted the issue of hydrological drought and its impact on agricultural, socio-economic and streamflow in the watershed (Azadi et al., 2018; Iqbal et al., 2016; Tigkas et al., 2012).

The hydrological drought was referred to as the most critical aspect of drought with significantly reduced streamflow and lower water storage in the river system (Hasan et al., 2019). Because of this, in order to ensure that water supply requirements are met, the storage rate for each river should be known to ensure that the minimum storage during low flow and drought in the coming years will be able to accommodate consumers' water demand. Some relevant research questions in the investigation of hydrological drought are: (1) 'Is there an increasing pattern in the streamflow in the Selangor region and is the streamflow trend the same throughout the year?'; (2) 'What is the likelihood of frequency of low flow conditions in the river system in the Selangor state?'; (3) 'What is the minimum required storage draft-rate based on monthly time series?'; and (4) 'How well does the threshold level method performs in determining the hydrological drought characteristics?'. The primary purposes of this study are: (1) to arbitrate the trend analysis of streamflow for 40 years; (2) to determine the best-fitted distribution of probability for each station for low-flow frequency analysis; (3) to determine the minimum storage draft rates in seven (7) catchments in Selangor region in Malaysia; (4) to evaluate the hydrological drought characteristics, including severity, duration and magnitude. This study is essential to understand the concept of low flow, drought characteristics, and the predictive significance of river storage-draft rates in managing sustainable water catchment. The findings are useful for designing strategies to sustain the variability of flow and can be used to implement risk management policies. Thus, this study consists of four types of analyses, which are: (1) daily streamflow trend analysis for a 40-year time series using the Mann-Kendall, Sen's slope, distribution-free (CUSUM) and Pettitt's test; (2) a low flow frequency analysis on annual minimum flow using the best fitted of distributions; (3) the determination of minimum storage draft rates necessary to ensure the sufficiency of

water supply during low flow periods; and (4) an analysis of hydrological drought characteristics determined using a fixed drought threshold at the 90[th] flow percentile.

## 2 Study area

The scope of this study covers the entire streamflow station in the Selangor state. Selangor covers an area of 8,104 km$^2$ and is located on Peninsular Malaysia's west coast. Selangor's water supply system not only covers the state of Selangor but also supplies water to the Kuala Lumpur and Putrajaya areas (Sakke et al., 2016a). Langat-, Klang-, and Selangor-River basin are the main river basins in Selangor. There are also three other river basins in Selangor which are the Buloh-, Bernam-, and Tengi-River basin. Table 1 shows the locations and characteristics of all streamflow gauging stations involved in this study. Langat and Semenyih dams, located at the upper reaches of the Langat river (Elfithri et al., 2018), serve to regulate the raw water supplied to treatment plants downstream. The main tributaries of Selangor rivers are Sembah, Kanching, Kerling, Rawang, and Tinggi river. There are two dams, namely Selangor and Tinggi dam, in the Selangor river basin.

Selangor state is characterised by its geographical position, which lies near the equator climate that is warm and humid over the year (Lassen et al., 2004). The average annual temperature varies between 27-30 °C, and the average annual relative humidity is between 70-90% (Lee et al., 2013). The equatorial climatic regions are influenced by two monsoons: the southwest Indian monsoon and the northeast Asian monsoon, which result in two rainy seasons with a significant amount of storm resulting in a mean annual rainfall of about 2500 mm (Mamun et al., 2010). Even though Selangor is located in the humid region, it occasionally encounters drought periods. Dry spells, low rainfall, and high soil impermeability due to population growth are the leading causes of low flow events. Low flow usually refers to a stream's regime that indicates the average annual streamflow variability associated with the regional climate's annual cycle. A stream's regime can display one or more low flow events depending on the climate. Two rainy and two dry seasons represent the equatorial climate, and the two streamflow regimes have two corresponding periods of high flow and low flow. Figure 1 shows the seven streamflow gauging stations involved in this study with four streamflow gauging stations located at Langat River basin at Dengkil, Kajang, Semenyih, and Lui. There is also streamflow gauging station at Rantau Panjang for the Selangor River basin, Tanjung Malim, and JAM SKC for the Bernam River basin, respectively (Department of Irrigation and Drainage Malaysia, 2011). The headwater of the Langat river basin starts from the northeast of the basin, flows to the southwest, and joins with the Semenyih River. The Langat and Semenyih dams, Selangor and Tinggi dams are located at the upper reaches of the Langat River and Selangor River basins, respectively, (Elfithri et al., 2018) to regulate the quantities of streamflow to the treatment plants.

# 3 Methodology

Daily streamflow data were obtained from the Department of Irrigation and Drainage Malaysia, which covers approximately 40 years (1978 to 2017) of records for all streamflow gauging stations. Precautions were taken to ensure reasonable low flow data were captured. The framework of methodology was developed for assessing the hydrological drought characteristics in the state of Selangor, Malaysia, using low flow and threshold indicator. The first analysis in this study is to determine the daily streamflow trend for 40 years using the Mann-Kendall test; and the slope of trend was calculated using the Sen's slope

estimator; the change points are identified using the CUSUM and Pettitt's test. Next, the potential of a probability distribution that optimally fits the 7-day mean annual minimum (MAM) in low flow frequency analysis was evaluated for determining different return periods. The 10-year return period was computed using the estimation of minimum storage draft-rate in the river using mass curve. Next, the threshold level was obtained from the flow duration curve (FDC), and 90th percentiles were selected for drought analysis. Finally, the characteristics of hydrological drought were analysed, including drought events,

durations and drought deficits in seven watershed catchments. The summary of the whole methodology analysis is depicted in Figure 2. The following sections elucidate the specific components incorporated into the methodology framework.

3.1 Streamflow trend analysis

The mean annual streamflow was analysed for significant trends, and distribution changes are discussed. The trend slope is measured using the Sen's slope estimator, that produces the magnitude of change in trends. Finally, using the CUSUM test,

the change points were defined in the long-term streamflow results, and the changes in streamflow before and after the change points were examined using the Pettitt test. All analyses were conducted in seven (7) stations to recognise the spatial variability based on historical streamflow pattern change. Mann-Kendall and Sen's T-tests are the most commonly used non-parametric trend analysis methods (Hisdal et al., 2001). Mann-Kendall test was chosen due to its capability of identifying the trend in a time series, if there is any. In the streamflow time series data, the trend was analysed using the Mann-Kendall test to evaluate

the significance of monotonic trends. For the test consist of a series of streamflow data over a time period, the null hypothesis ($H_0$) is tested, and the data originates from a series of variables that are identically distributed and independent. The data of $H_1$, the alternative hypothesis, follows a monotonic pattern over time. Under $H_0$, the test statistics for Mann-Kendall are given by Eq. (1):

$$S = \sum_{i=j}^{n-1} \sum_{j=i+1}^{n} sgn(x_j - x_i) , \tag{1}$$

where $x_j$ and $x_i$ are the data values in years $j$ and $i$, respectively; and $n$ is the total number of years. The probability associated with $S$ and the sample size, $n$, is determined to measure the trend significance statistically. The normalised test statistics, $Z$, is expressed as follows using Eq. (2):

$$Z = \begin{cases} \frac{S-1}{\sqrt{VAR(S)}} & (S > 0) \\ 0 & (S = 0) \\ \frac{S-1}{\sqrt{VAR(S)}} & (S < 0) \end{cases} \tag{2}$$

The null hypothesis of no trend is rejected if $Z > 2.575$ at 99% significance. In the test statistic, $S$ calculates the sum of the difference between data points and the associations between samples to show the presence or absence of a trend. When the value of $Z$ is positive, it gives a positive trend, and a negative trend when $Z$ gives a negative value. In this study, the level of significance of 0.05 or 95% ($P$-value = 0.05) was used. If their $P$-value was equal to or less than 0.05 ($P$-value $\leq$ 0.05), the trend test is considered significant, as shown by Eq. (3) (Coch and Mediero, 2016):

$$Trend = \begin{cases} + & (Z > 0) \\ 0 & (Z = 0) \\ - & (Z < 0) \end{cases} \tag{3}$$

Then, a linear trend analysis was also conducted, and the trend magnitude was determined using the Sen's slope method. Sen's slope is a non-parametric method for determining any trend's slope. It utilises data from a time series that is similarly distributed. The difference in slope was calculated per changed time for each data point. If a trend is identified in a time series, the slope can be determined using the slope estimator ($\beta$) in Sen's slope test. For the entire data set, the estimator, $\beta$, is the median of all slopes between data points. A positive $\beta$ indicates an increasing trend, and a negative $\beta$ indicates a decreasing trend as given by Eq. (4):

$$\beta = \text{Median} \frac{y_j - y_i}{x_j - x_i}, \tag{4}$$

with $n$ the number of data; $i, j$ are indices with $i = 1, 2, \ldots (n\text{-}1)$ and $j = 2, 3, \ldots, n$. The changes in the average annual streamflow were determined after the trend slope has been verified, using the equation employed by Petrow and Merz, (2009) to calculate the amount of change in the data series by Eq. (5):

$$\Delta X_R = \frac{X_{end} - X_{first}}{X_{mean}}, \tag{5}$$

where $\Delta X_R$ is the amount of change observed in the data series, $X_{end}$ is the last piece of the trend slope data, $X_{first}$ is the first piece of the trend slope data, and $X_{mean}$ is the mean of all piece of the slope. The distribution-free CUSUM test is a cumulative total of time series deviations of target value and is capable of detecting abnormal trends, simplicity and better graphical representation of results (Sonali and Nagesh Kumar, 2013). Let us consider $x$ samples, each of n size with mean $\mu_0$ and standard deviation $\sigma$. Then, the cumulative sum of deviation ($S_i$) from the target value (mean) was calculated using Eq. (6):

$$S_i = \sum_{j=1}^{i}(x_j - \mu_0), \tag{6}$$

where $x_j$ is the mean of $j$th sample. Finally, by considering a sequence of random variables $x_1, x_2, ..., x_T$ which may have a change-point at $N$ if $x_t$ for $t = 1,2,..., N$ has a common distribution function $F_1(x)$, the Pettitt test index ($U$) is defined as Eq. (7) (Ahn and Palmer, 2016):

$$U = \sum_{i=1}^{T} \sum_{j=T+1}^{n} sgn\,(x_j - x_i)\,, \tag{7}$$

Where, $T$ = change point, $x$ = target variable and $sgn(x_j - x_i)$ is defined as Eq. (8):

$$sgn\,(x_j - x_i) = \begin{cases} +1, x_j > x_i \\ 0, \ x_j = x_i \ , \\ -1, x_j < x_i \end{cases} \tag{8}$$

The non-parametric statistic (Eq. 9) was applied in the evaluation of change point at which time $U$ has the highest absolute value.

$$K = Max_{t \le T \le i}(U)\,, \tag{9}$$

where $K$ = final Pettitt statistics and $T$ = data point at which the change occurs. The probability of significance was approximated by $p \approx 2 \exp\,[-6K^2\,(i^3 + i^2)]$. When $p$ is smaller than the specified significance level (0.05), the null hypothesis is rejected.

### 3.2 Low flow frequency analysis

There are many types of frequency distribution function that have been applied successfully to hydrological data. Frequency analysis is based on fitting the observed data with a theoretical probability distribution function and providing low flow estimates for any given return period. The choice of probability distribution is defined as the distribution of probability with the shape parameter. This selection is necessary to evaluate the shape parameter as the parameter for skewness. The frequency analysis starts with the calculation of the annual 7-day minimum streamflow series for each gauge station in order to determine

the suitable probability distribution that best fits the minimum 7-day low flow in Selangor. Then, four probability distributions, including the Gamma distribution, Gumbel, Lognormal 2P and Pearson type 3 distribution (PE3) were evaluated to determine which distribution most appropriately fits the low flow data. The Kolmogorov–Smirnov (K-S) test and ranking method were used to determine the best fitting distributions. After choosing the optimum probability distribution, it is important to estimate the return values for certain return periods. The return period of low flow occurrence is crucial for determining the magnitude

and frequency of low flow, and such information is useful in minimising and mitigating the risk of drought in future. Four scores ranging 1 to 4 represent the ranking of distributions in fitting the data, were assigned to each station, where score 1 indicated the best while score 4 indicated the worst. The summation of scores shows the suitability of distribution such that the best distribution got the lowest sum of scores. The selected regional probability distribution function was then used to calculate the annual 7-day minimum discharge series with a 1-, 2.3-, 5-, 10-, 25-, 50-, and 100-year return period. The 7-day

minimum with a 10-year return period (7Q10) was used to derive the minimum storage-draft rate required for all stations (Section 3.3).

The probabilistic behaviour was analysed using four probability distribution functions (PDFs), widely used in extreme value analysis (Joshi and St-Hilaire, 2013; Zaidman et al., 2003). Then, probability distribution functions were fitted with their
parameters estimated using the method of maximum likelihood estimation (Assefa and Moges, 2018). Goodness-of-fit was determined by the Kolmogorov–Smirnov test. Here, a 95% confidence level was accepted to reject or accept a non-reject hypothesis, based on $D$-value. The graphical illustration of probability plot is described as the $i^{th}$-order statistic of the sample, $y(i)$, as a function of a plotting position, which is simply a measure of the non-exceedance probability related to the $i^{th}$-order statistic from the assumed standardised distribution (Sharma and Panu, 2015). The $r^{th}$-order statistic is acquired by the way of
rating the observed sample from the smallest ($i = 1$) to the greatest ($i = n$) value, then $y(i)$ equals the $i^{th}$ largest value. According to Koteia et al. (2016), the plotting position of low flow, $P$, can be obtained using the Weibull formula given by Eq. (10) (Koteia et al., 2016):

$$P = \frac{m}{(N+1)} , \qquad\qquad (10)$$

where, $P$ = The probability of low flow; $m$ = the ranking, from highest to lowest, of mean annual minimum flow; and $N$ = the
total number of the mean annual minimum flow. The probability selection is made following the shape parameter. This is because it is possible to represent the shape parameter as the parameter for skewness. For each distribution, Table 2 provides the functions of probability density. For this study, the method of maximum likelihood is used for parameter estimation. The likelihood function is defined in Eq. (11):

$$l(\theta|\, x_1, x_2, \ldots, x_N) = \prod_{i=1}^{n} f(x_i : \theta_1, \theta_2, \ldots, \theta_N) , \qquad\qquad (11)$$

Once the parameters are estimated, the selected distributions will be tested for the assumption that the observed data is actually from the fitted distribution of probability. The Kolmogorov-Smirnov (KS) test has been used to determine the largest discrepancy between the theoretical ($F_n(x_i)$) and empirical ($F_0(x_i)$) cumulative distribution functions. The KS test obtains a $D$-statistic; the maximum vertical is given by Eq. (12):

$$D = \max\left(|F_n(x_i) - F_0(x_i)|\right) , \qquad\qquad (12)$$

Where $r$ is the rank of the observation, $i$, in ascending order, the smaller $D$-values imply a better fit of the streamflow series to the selected probability distribution. If $D$ was higher than the critical value ($\alpha = 0.05$), the distribution was rejected. After the probability calculations, $P$, and subsequent returns period the low flow, $T$, the low flow rate variation will be plotted against the return period, $T$ on the semi-log graph. With this graph, the specific magnitude of a specified period can be determined (Erfen et al., 2015; Gottschalk et al., 2013). The return period in a univariate setting is described in Eq. (13):

$$T = 1/(1 - P) , \qquad\qquad (13)$$

Where, $T$ = the return period (year); $P$ = the non-exceedance probability.

## 3.3 Minimum storage-draft rate method

The water supply or inflow is depending on low flow characteristics in the stream. If the inflow rate is lower than the outflow (demand) rate, the cumulative difference between supply and demand volume is the maximum amount of water drawn from storage during the dry season. In channel storage, the function of both outflow and inflow discharge can be considered under two categories as prism and wedge storage. The water surface flow in the channel is not only unparallel to channel bottom but also varies with time. The storage, which is the maximum cumulative deficiency in any dry season, is obtained from the maximum difference in the ordinate between the mass curve of water supply and demand. Thus, the storage required can be expressed as per Eq. (14):

$$S = Maximum\ of\ (\Sigma V_D - \Sigma V_S), \tag{14}$$

Where, $V_D$ = Demand Volume; $V_S$ = Supply volume.

The minimum storage draft rate was determined by using the mass curve of low flow at a monthly interval (Bharali, 2015). Although specific evaluation of storage requirements is essential for design, reconnaissance planning can frequently be facilitated by using draft-storage curves based on low flow frequency analysis. Alrayess et al. (2017) determined the capacity of river storage by the mass curve method. The mass curve has many useful applications in the design of storage capacities, such as to determine the storage capacity and flood routing (Gao et al., 2017).

The mass curve method can be used to define the storage required for a given draft-rate for monthly of record. This approach is limited to draft-rates that can be sustained by the streamflow available in any one month; that is, by within-a-year of storage. The usefulness of this analysis depends on the monthly variability of streamflow. In some regions, the maximum draft that can be provided is less than a tenth of the mean flow. In others, notably in Selangor, drafts of half of the mean flow can be provided by within-a-year of storage. The estimation of the storage draft-rate in this study will determine the minimum storage of a river to sustain the water supply during low flows and droughts. The mass curve of the monthly low flow rate is used in this analysis to obtain the minimum storage rate of the river. The procedure for the mass curve method has the following steps; first, the mass-curve analysis of low flow for the duration of January to December was plotted against duration for recurrence interval of 10-year from 10 years return period in Table 7. Second, the cumulative draw off that corresponds to a constant draft rate of 50% of the mean annual flow and was connected by a straight line. Third, the cumulative draft line was superimposed on the mass curve; fourth, the largest intercept between the cumulative draft line and the mass curve was measured. The maximum positive difference between cumulative draw-off and low flow is the minimum storage necessary to maintain a draft-rate of 50% of the mean annual streamflow. The example of minimum storage required in the river for station S05 using mass curve analysis was shown in Figure 3.

## 3.4 Threshold analysis

An approach based on deficit characteristics under a given threshold method was adopted to identify extreme low flow occurrences (Fleig et al., 2006). The low flow period, which depends on the catchment's hydrological regime, is defined by a fixed threshold level. The selection of the threshold level is influenced by the study objective, region, and available data. The threshold level method can easily obtain the start and the end times of drought or streamflow deficit period and has been used to define streamflow droughts or deficits. The fixed threshold level in this study is the 90th percentile value (Q90) of FDC, which was compiled using all available daily streamflow and identified as perennial rivers with river flow having continuous flow.

The low flow value was obtained from the flow duration curve at 90[th] percentiles. Flow Duration Curve (FDC) describes the ratio of a specified percentage of time with discharge being equal to or surpassed (Croker et al., 2003; Mohamoud, 2008; Vogel and Fennessey, 1994), which reflects the relationship between streamflow magnitude and the length of time that relates to the average percentage of time of a specific flow is exceeded (Sung and Chung, 2014). The FDC was developed by arranging streamflow values in decreasing magnitude order and assigning rank numbers to each streamflow value. The most substantial flow was ranked as one, and the smallest flow was ranked as $n$, where $n$ is the complete record quantity. The percentage of time for a given flow was equal to or exceeded (probability of excess) when calculated using the relationship in Eq. (9) (Awass, 2009; Koteia et al., 2016; Yahiaoui, 2019):

$$P = [r/(n + 1)] \, X \, 100, \tag{9}$$

where, $P$ = the percentage of time a given flow is equalled or exceeded; $n$ = the total number of records; $r$ = the rank of the flow magnitude. Kannan et al. (2018) indicated the flow duration curve could be divided into five zones, representing high flows (0-10%), humid conditions (10-40%), medium-range flows (40-60%), dry conditions (60-90%), and low flows (90-100%). The selection of percentile will strongly condition the classification and evaluation of extreme low-flow events. The magnitude of drought characteristics was determined by the threshold value and difference in value between the time series. When compared to the use of standardised drought indices, a major benefit of this approach is that it allows the deficit volume to be quantified, which is a critical aspect in the management of water supplies. When the flow falls below the threshold level, a drought event begins and terminates when the flow exceeds the threshold level. The duration, total deficit which is the sum of the deficits, and magnitude of each drought event can be readily obtained. As the daily data series was used, the existence of minor drought events and mutually dependable drought events can be detected (Van Loon and Van Lanen, 2013). In order to deal with this problem, pooling procedures such as moving average, inter-event time criterion and inter-event time and volume criterion were frequently used (Sung and Chung, 2014). According to the study by Sakke et al. (2017), to eliminate the minor drought events, the events that have occurrence of less than 15 days will be excluded while the mutually dependable events were also eliminated by the pooling procedure (Sakke et al., 2016b). In this paper, the 7-day moving average was applied as a pooling procedure to obtain smooth data. Through these methods, the mutually dependent drought events will

combine into individual and independent drought events (Fleig et al., 2006). The minor drought events will be eliminated or combined with individual drought events automatically (Yahiaoui et al., 2009).

## 4 Results and Discussion

The streamflow data from the seven streamflow gauging stations will be analysed in three aspects, which are mean annual low flow and the probability of occurrence, drought characteristics using the threshold level and the estimation of storage draft rate of the river. Statistical characteristics were calculated from the observed 40 years daily streamflow time series: the mean, minimum, and maximum; standard deviation, skewness, and kurtosis for each station (Table 3).

### 4.1 Streamflow trend analysis

Annual streamflow series trend analysis presents the overall view of the shift in systems of streamflow (Assefa and Moges, 2018). The Mann-Kendall test, Sen's slope, relative change within 40 years, maximum cumulative sum (CUSUM) with the year of change point and their value of $p$ using Pettitt test are displayed in Table 4. In trend significance test, the significance level of $\alpha = 0.05$ was set as the standard, making $Z_{\alpha/2} = 1.96$. The analysis indicated that five selected stations (S01, S02, S04, S05, and S07) have increasing trends of streamflow. Two of the stations, S03 and S06, showed a decreasing trend with the negative change of streamflow. The estimation of trend slope was carried out using the Sen's slope estimator, where an upward (downward) streamflow trend is indicated by a trend slope greater (less) than zero. In order to compute the trends of annual streamflow, the trend slope values were also used to construct a trend line. Using Eq. (5), the amount of change in annual streamflow was determined. The analysis results indicate that the amount of change in the basin of station S04 was higher than that of at other stations (Table 4). The two gauging stations, which are S03 and S06, had significantly greater changes that showed a downward decreasing trend of -20% and -55%, respectively. Streamflow trends indicate variability from one station to another, in terms of magnitude and trend direction. This variability resulted from several factors, due to potential human intervention or change in environment at regional bases. In the S03 and S06 stations, there could be several factors for decreasing streamflow. Some of this involves modifications in the catchment of physical characteristics such as changes in land cover in river basins (Hisdal et al., 2001). Another five stations indicated an increase in trends of streamflow due to climate change for the increasing temperature and soil water evaporation (Siwar et al., 2013; Taye et al., 2011).

The accuracy of the results of data analysis is of crucial importance in the trend analysis studies, especially on the discharges of any stream. The majority of station trends on the main and secondary branches of the basin reflected good consistency in this analysis. Two main rivers, however, demonstrate a paradox, although one station shows a declining trend and the other station shows an increasing trend. Due to the location of the stations, dam construction, link of another stream to the channel, irrigation and other disruptions in the discharge regime of the river, this condition is foreseeable. Stations S01, S02, S03 and S04 are located on the same stream, but the trends at station S04 are not in the same direction. Stations S01, S02 and S03 have

a significantly increasing trend while station S04 shows no significant downward streamflow trend, caused by the disruption in the river regime, such as the construction of a Langat Dam, may cause this contrast (Memarian et al., 2012).

The results of the change point in annual streamflow are tabulated in Table 4 using the Pettitt test. For each time sequence, the result gave the most likely change point event. For the annual streamflow, the results showed that 1997 was the most probable year of change with a p-value = 0.0004. Some stations show signs of change point at a significance level of 5% while the others do not. The prediction of process changes and trend generation are well indicated using CUSUM charts. This analysis shows a change point that can be seen in the year of 1996, with a confidence interval setting of 95%, and the *p*-value of 0.1215

for station S01. The change point occurred in 2005 twice for station S05 and S07 in Selangor state. The major changes in the annual streamflow observed revealed that the presence of rapidly increasing industrial activities in the basin due to a shift in the land use is caused by the result of the streamflow trend in the basin. The latest change points occurred in 2009 at Bernam River (S06) with new implementation of several projects by the state government such as the construction of feeder canal for agricultural and repairing of the collapsed stretch of the riverbank caused the widening the river channel.

For the mean annual streamflow at the gauging stations, five stations indicated an upward trend, and two stations indicated a downward trend for a 40 years' data. The interpretations of trend analysis for relatively partial streamflow records may only reflect a short-term condition and may not be a representative of an actual long-term change in the streamflow data. This issue is valid for relatively short-term records that begin or end in a historically low flow condition. From the average annual

streamflow results, the change point is seen to be present at a 100% confidence interval in 1996-1997 and 2005-2007, and implies that there is an impact of rapidly increasing industrial activities in the basin as well as a change in the pattern of land use induced by the effect of streamflow patterns in the basin. This study is very useful in interpreting climate change scenarios and is focused on the revealed characteristics of regional-level hydrological variables.

The anthropogenic has taken place in transformations of water surface such as the construction of reservoirs, trans-basin diversion project, crop irrigation, urban water supply or drainage, and urbanisation. There are three strategic dams in the study area. Those are Langat Dam in S02, Semenyih Dam in S03 and Sungai Selangor Dam in S05. All dams are functional for domestic and industrial freshwater supply. Whereas, the Langat Dam is only used as a power supply generator for the Langat Valley consumption. A study by Shaaban and Low (2003) showed that drought events reduced water discharge at the Langat

and Semenyih basin, particularly in the period of 1993–1998 (Shaaban and Low, 2003). This event justified the change point from this analysis. These drought events have decreased the trend of water discharge in the Semenyih basin. Due to the increasing size of natural or artificial dams, the reduction of streamflow trend was regulated at the Langat river basin as compared to the Semenyih basin.

Streamflow variability due to potential human intervention or climate change is important for regional water supply planning and management. Knowledge of streamflow variability and its trend is crucial for the socio-economic sector because any changing in streamflow is a limiting factor for the use of water resources. The streamflow decreasing trend, could result in important economic losses and affect health and human welfare, as well as the aquatic ecosystems. One of the influential aims of the time series trend is to define the nature characteristic represented by the sequence of observations and predicted future values of the time series variable. The analysis of the observed data for changes and trends of streamflow data can be used to assess the impact of climate change. The streamflow trend can estimate future water availability to maintain and sustain ecosystem functions. Moreover, streamflow trend analysis can also be used to predict any change in river flows for making water withdrawal decisions, which indirectly could improve drought management response.

## 4.2 Low flow frequency analysis

Frequency analysis has focused on fitting a theoretical probability distribution function to the observed data and providing low flow estimates for any given return period. For each station, annual minimum streamflow was plotted using all the distributions. The goodness of fit was performed using Kolmogorov-Smirnov. All the PDFs were ranked for streamflow at each station. Ranks, according to this three goodness of fit, showed a significant variation. In the case of annual minimum streamflow, various distributions were found to be the best fit for different stations, namely, Gamma, Gumbel, Lognormal 2P and Pearson type 3. Figure 4 shows the example probability of mean annual minimum flow for station 1. The estimated parameters were determined and shown in table 5. The information on the return period of extreme events can be used in determining the risk management by extreme events such as hydrological drought, while the geographical station location and the surrounding environmental factors for the variation of streamflow. Table 6 shows the best-fit results of the K-S test and P-value results with their ranking.

The purpose of the probability distribution fitting is to represent the low flow probability most accurately. Among all stations, it was found that among all distributions, the Lognormal 2P yielded the most cases of best-fit distributions, while the Gumbel and Gamma yielded the second and third amount of best-fits, respectively. Comparatively, it is proposed that Lognormal 2P distributions predict low flow discharges for all the rivers under analysis, which can be used in water quality and quantity management at gauged and ungauged areas. From this comparison, although 3-parameters in the probability distribution functions are more advantageous to fit the 7-day low flow sequences better. However, in Selangor region, 2-parameter is more suitable which optimally fits to a 7-day mean annual minimum flow verified in the studies of Granemann et al. (2018) and Lelis et al. (2020). When the best fit probability distribution of the low flow series of the 7-day has been determined, the low flow discharge of the 7-day can be estimated according to any given return period. It should be noted that the research is station dependent on this analysis. Table 7 shows the return period of low flow at all streamflow stations. The 7-day mean annual minimum for recurrence interval of 10-year (Table 7) was used in the determination of minimum storage draft-rate for each station.

A catchment with a slow or quick response to rainfall intensity that usually has prolonged or rapid recession actions depends entirely on the catchment's physical characteristics. Low flow in catchments that respond quickly is lower than in those that respond slowly. Low flow in catchments that respond slowly is more persistent than in catchments that respond quickly. These differences demonstrate the significant effect of hydrological processes and storages to the low flow events. Figure 5 displays the low flow relationship with the watershed area represented by the boxplot graph. The largest range for low flow per area is in S06 while the smallest range is in S01. The boxplot graph provides information about the shape of a data set. S01, S02, and S04 are skewed right; S03, S05, and S06 are symmetrically shape data, and S07 is skewed left. From the discussions above, it is clear that the natural elements that affect a variety of factors of the river's low flow regime consist of distribution and hydraulic components, climate, and topography.

## 4.3 Estimation of minimum storage draft-rate

This study focused on the minimum surface water storage required based on the records from the hydrological stations in the Selangor state for the 1978 to 2017 period. Hydrological drought is a recurring phenomenon of water shortage that incorporates the storage of surface and subsurface water under the effects of climate change and human activity (Schwalm et al., 2017). The water storage required for all stations is based on their respective monthly streamflow discharge. A graph of cumulative streamflow draft-rate versus a specific historical timeline is plotted to find out the storage required of each station. Figure 6 shows the mass curve analysis for the determination of minimum storage-draft rate of each station that needs to be maintained at a draft rate of 50% of the mean annual flow during low flows to sustain the water supply.

The minimum storage required for maintaining a draft rate required for S01 is 21.51 $m^3$/s in October, S02 is 13.37 $m^3$/s in December, S03 is 4.79 $m^3$/s in December. The minimum storage required for S04 is 2.32 $m^3$/s in October for a 40 years' duration period; S05 is 15.00 $m^3$/s in September. While, the minimum storage required to maintain the draft rate for S06 is 10.90 $m^3$/s in October, and lastly, for S07 is 6.17 $m^3$/s in September. The result shows the water storage for all stations did not meet the corresponding water required, while stations S05 and S07 correspond to the required expectation for August to October. This result reveals that the September to December period is a critical duration in river water storage to sustain the water availability during low flow in a 10-year occurrence interval. This finding is justified by Selangor state located at the west coast of Peninsular Malaysia which is affected by two main monsoon seasons and two inter-monsoon seasons with October and January being relatively dry months (Hazir et al., 2020). However, there is not enough water storage starting September for station S05 and S07.

Low flow and surface water storage assessment is a critical issue for understanding the global water cycle, which is recognised to be of significant importance on a regional and global scale for the monitor of water resources. Correspondingly, this analysis provides important scientific data on the minimum storage required for river systems. Sufficient water storage during critical

dry periods is largely dependent on the adequacy and efficiency of water supplies from surface water resources. This surface water storage faces many challenges, which could lead to a decrease in their optimum yields and eventually leading to an inadequate supply of water over the next ten (10) years. This could be due to reasons such as increasing water demand due to increasing population and industry needs; and emerging demands for recreation and the conservation of the quality of stream water, biodiversity, and aquatic ecosystems.

## 4.4 Hydrological drought characteristics analysis

The threshold level value per Q percentile obtained from the flow duration curve is shown in Table 8. In this study, only $Q_{90}$ was used as a threshold level in the determination of drought events. The percentage where the streamflow rate was below the average level and the respective days were recorded to show the severity of droughts events at each station. The growing perception of hydrological drought improvement on a global scale has some necessary implications for water management. It is recognised, for example, that the duration and the volume of the deficit of the drought are associated (Fleig et al., 2006). Figure 4 to 7 show the drought characteristics below the threshold level ($Q_{90}$), with the minor drought for each station in the Selangor region removed.

Station S01 has 39 episodes of drought events in 40 years. This station also recorded 1593 days of drought, with a total deficit of 10,299.97 $m^3$/s. The lowest deficit was recorded in 1994 at 41.53 $m^3$/s, while the highest deficit was recorded in 1986 at 666.58 $m^3$/s. The average amount of water deficit was 264.10 $m^3$/s. This river has been affected by water rationing that happened in Selangor in early 2014 for 3 to 4 months. The most prolonged period of individual drought was recorded in 2014 at 112 days from March 05 to June 24. The shortest period of a single drought was 15 days, which was marked three times in 2004 and 2005. Station S02 was a part of the Langat river basin and has had 29 episodes of drought events in 40 years. The total duration of the drought events was recorded to be 1,261 days from the 14,610 days of total observation, which was only 8.63% of the entire record period and was below the threshold level Q90 = 2.99 $m^3$/s. The overall deficit for this station was 2,340 $m^3$/s, with an average of 80.70 $m^3$/s. The lowest deficit was in 1993 at 34.44 $m^3$/s, while the highest deficit was recorded in 1986 with 179.73 $m^3$/s. The overall total deficit was 1.57% of the total water flow.

The threshold level of S03 was 1.47 $m^3$/s at an average level with 12 episodes of drought events. The total number of the occurrence of drought was 1,577 days, which was 10.79% of the overall record of observation. S03 has the lowest record value of the total number and series of drought events among all stations. However, S03 also recorded a long period of drought for individual events. The longest single drought took place in 1998, with 241 days commencing on February 24 and ending on October 22. S03 also recorded the lowest deficit amount amongst all stations with 1,660 $m^3$/s during the period of drought. This total was 2.2% of the total water flow through this station, which was 75,562 $m^3$/s. The highest deficit was recorded in 1998 with a total of 226 $m^3$/s over 241 days. The lowest deficit was recorded in the dry season in 1997, with only 21.57 $m^3$/s within 20 days. Station S04 has 28 episodes of drought occurring in 40 years of records. The most prolonged period of

individual and annual drought was recorded in 2004 by 306 days. The shortest period was 15 days in 1999. The number of
drought events exceeding the number of years of drought was due to repeated events occurring 18 times with a maximum of
four (4) replications in one (1) year. The total number of days of the occurrence of this drought was 1,460 days, which is 9.99%
of the total daily flow data. The overall deficit of 28 drought events was 673.54 m³/s. The lowest total deficit was recorded in
1983 as much as 7 m³/s, while the highest deficit was recorded in 2004 with 131.27 m³/s. The average amount of total deficit
was 24.06 m³/s.

Station S05 has been categorised as the most critical station with the highest number of days of droughts events. The longest
annual drought event was recorded in 1998 with 217 days, and for individual drought events, this occurred in 1999 with a
period of 111 days. Using the threshold level at Q90 = 21.52 m³/s, 1,236 days (10%) of the total are below the threshold level
categorised as drought. Repeated drought events were recorded in 1978, 1979, 1986, 1987, 1990, 1998, 2000 and 2002. The
drought episode was seen most repetitive in 1998 with four (4) repetitions a year. The total magnitude deficit of the entire river
water stream during the occurrence is 18,695.45 m³/s. The value of minimum storage rate at 67.36 m³/s exceeds the amount
of low flow rate at 35.61 m³/s that will occur at a return period of 50-year. Station S06 shows the drought episodes were seen
in succession from 2011 to 2017 and 2016 recorded the highest drought events with four (4) replay events. The year 2014
recorded the most extended individual drought episode of 177 days, and the longest annual drought came in 2013 with 372
days. S06 recorded a total deficit of 3,847 m³/s. The year 2012 recorded the highest deficit of 496.13 m³/s while 1989 recorded
the lowest deficit with only 54.19 m³/s. The average deficit was 113.16 m³/s, with 34 episodes of drought event in 40 years.

S07 had the highest drought events with the number of years of drought recorded as 39 years with repeated drought events in
1978, 1983, 1985, 1987, 1990, 1991, 1992, 1998, 1999, 2001, 2002, 2005 and 2016. The most prolonged drought period was
recorded in 2005 with a period of only 99 days, while the shortest period in 1971, 1987, 2000, and 2016 with a period of 15
days. The most prolonged period of individual drought events with 205 days occurred in the same year in 2005. The total
drought days at this station was 1,614 days, which was 11.05% of the total days. S07 recorded a deficit of 21,740 m³/s during
the drought episode, and this percentage is the highest percentage recorded as compared to other streamflow stations. This
stream recorded a high deficit amount with fewer drought days. The highest deficit reached was 1,445 m³/s, which was
recorded in the drought events in 1990, while the lowest deficit was in 1983 with a total of 161.32 m³/s.

From the results, S01 exhibits the highest number of drought events, at 39 episodes, with the mean deficit being 264.10 m³/s.
This station is located downstream of the Langat basin. It indicates the downstream watershed catchment has more drought
episodes compared to the upstream catchment. Magnitudes differ significantly between catchments since there were also varied
specific hydrological characteristics, such as station spatial distribution, precipitation and temperature magnitudes, and
frequency of extreme events like drought.

Several indices could be used to provide a more accurate representation of hydrological drought. Which indices one chooses to use is going to affect the result directly. It is important to note that the $Q_{90}$ threshold merely identifies low flows accounted for catchments regular flow, especially in this study area. Therefore, the $Q_{90}$ threshold does not necessarily imply a situation where functions in nature are affected. The threshold level can reflect a specific requirement, such as for water supply or minimum environmental flow, or a normal low flow condition of the river can be represented. For a bigger picture and understanding of the broad spectrum of hydrological drought, more indices need to be put together in an index. Different methods will allow different characteristics of hydrological droughts. The threshold level method should be used for more detailed deficits and in-depth study. Complex indices would be most useful to verify results in regional studies. While streamflow changes are mainly influenced by rainfall variability, the occurrence of low flow conditions is also likely to be a function of catchment response, influenced by catchment storage. There can be a significant variance in the frequency, severity and duration of streamflow depletion between surrounding catchments as a drought develops and subsequently decays. In catchments with low storage, streamflow levels typically drop more rapidly than in catchments that receive a consistent flow from stored sources. However, catchments dependent on stored water are becoming increasingly vulnerable in a prolonged or multi-year drought as depletion in groundwater storage begins to affect baseflow levels. Thus, even after rainfall has returned to normal levels, flows in permeable catchments may still be affected.

Selangor's river flow trend reflects the rainfall pattern, and there is a prompt response to rainfall in general, although the response rate varies from catchment to another. Some catchments, with little or insignificant storage, have a very rapid response to rainfall and are known as flashy catchments. The rate of increment in runoff resulting from rainfall in other catchments may not be as extreme as water goes into storage and then contributes to the flow of rivers from storage. Selangor State enjoys a tropical rainforest climate with two major monsoon seasons and two inter-monsoon seasons. Due to this, heavy rainfall typically occurs in the form of convective rains and the state is generally wetter than other parts of Malaysia Peninsular. Drought in Selangor is therefore not a very frequent event. However, not to forget, droughts events occurred in the past: 1986, 1994, 1997, 1998, 2003 and 2004 for all stations. This pattern justified the El Nino events that largely influence the climate variability over Malaysia, especially the Selangor state (Tangang et al., 2012). This situation can be seen with the drought period being very closely related to the amount of deficit that occurs. Drought is seen as very severe when it occurs over a long period, and the amount of water deficit experienced is a high.

## 5 Conclusion

This study determined the streamflow trend analysis on seven stations in the state of Selangor, Malaysia, to quantify the trends over 40 years of record data. The result shows that two stations experienced significant decreasing trends, with 55.56% of relative change within the 40 years. From the mean annual streamflow data, it is seen that the change point is present in 1996-1997 and 2005-2007 at 100% confidence interval and implies that there is an influence of fast-growing industrial activities in

the basin and there is also a change in land use pattern, which is caused by the effect of streamflow trends in the basin. This finding has important implications for water resources management, which will affect future developments in Selangor. The impact of serial and spatial correlation on the trends needs to be investigated. Further study in streamflow trends needs to be carried out, such as the prediction or modelling in the forecasting of streamflow trends.

Low flow analysis is an essential and widely studied design and management of hydrology and water resources. Varying and complex natural processes may produce low flows in a river on a catchment scale. The second aim of this work was to determine the characteristics of low flow by using frequency analysis. In order to determine the suitable probability distribution that optimally fits the minimum 7-day low flow values, first, the 7-day mean annual minimum streamflow series for each gauge was computed. Then, four probability distributions, including the Gamma distribution, Gumbel, Lognormal 2P and

Pearson type 3 distribution (PE3) were evaluated to determine the distribution that most appropriately fits the low flow data. The Kolmogorov–Smirnov (K-S) test and ranking method were used to determine the best fitting distributions. Based on the result, Lognormal 2P distribution provided a good fit to annual minimum flow data at each station. After the suitable probability distribution was selected, the return values for certain return periods were estimated. The return period of low flow occurrence is crucial for determining the magnitude and frequency of low flow, and such information is valuable in accessing and

mitigating the drought hazard in future. Their parameters define distributions of probability, hence, to better understand the theoretical probability distribution method, it is necessary to fully understand the principles underlying parameter estimation for established theoretical frequency distributions. From the result, the range indicated that the low flow of rivers in Selangor was between 0.75 to 19.47 $m^3$/s. The 7-day mean annual minimum for recurrence interval of 10-year was used in the determination of minimum storage draft-rate for each station.


The draft-rate of low flow at the recurrence interval of 10-year from low flow frequency analysis using Lognormal 2P was used to ensure the minimum storage draft-rate required to sustain the water demand during low flow periods. The restructuring of minimum storage draft rate must be carried out by hydrologist at a particular return period to ensure the streamflow gauging station has enough water to be supplied to the user during the low flow and drought periods. Based on the analysis of the study,

the estimated minimum storage-draft rates for each station cannot meet the water demand during low flow at specific return periods, which is a 10-year recurrence interval for this research. This result reveals that September to December is a critical period in river water storage to sustain the water availability during low flow in 10-year occurrence interval. The storage of river water faces several problems that may lead to a decrease in its sustainable yields and even to an inadequate supply of freshwater over the next ten (10) years.


Hydrological drought is a phenomenon of water shortage when the water supply is below the average level. This study developed a sound principle of using threshold level methods to describe the characteristics of streamflow droughts. However,

the threshold selection should be further analysed because it is not clear if Q90 should be used as a representative threshold for rivers in a tropical climate. From this study, we can make the following conclusions:

1) The threshold level using the Q percentile based on the flow duration curve was used as an average level to separate the occurrence of droughts events or otherwise. The number of days and duration of droughts for a station can show the severity of the drought that occurs.

2) The drought characteristics were analysed from time-series below a threshold level (Q90) with removing the minor drought. The magnitude and duration of drought characteristics were determined by the value difference between the
time series and the threshold level value.

3) The highest drought events are 39 episodes with a mean volume of the deficit being 557.46 m$^3$/s while the lowest events of drought were ten (10) episodes with the mean volume of the deficit being 127.71 m$^3$/s.

4) Drought in Selangor is therefore not a very frequent event. However, several notable droughts occurred in Selangor in the years of 1986, 1994, 1997, 1998, 2003 and 2004 for all stations.


This research is essential to water resources management. Low flow analysis and water availability enable water resource management to make more realistic decisions on water restrictions and provisions for cities and populations. Understanding the concept of low flow and the predictive significance of river minimum storage draft-rate required can also help in managing sustainable water catchment. This study also helps in emphasising the natural flow of water to provide water supply for
continuous use during low flow. Additionally, through this research, the concept of low flow analysis, hydrological drought using threshold level and the predictive significance of minimum storage draft rate can be developed to produce more efficient water resource management systems during the dry season in Selangor, Malaysia.

*Competing interests.* The authors declare that they have no conflict of interest.


*Acknowledgements.* The authors are thankful to the Ministry of Education Malaysia for the financial support of this research through research grant number FRGS/1/2018/TK01/UKM/02/2. The authors would also like to acknowledge their gratitude to the Department of Irrigation and Drainage (DID), Malaysia, for providing streamflow data for this study.

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

**Figure**

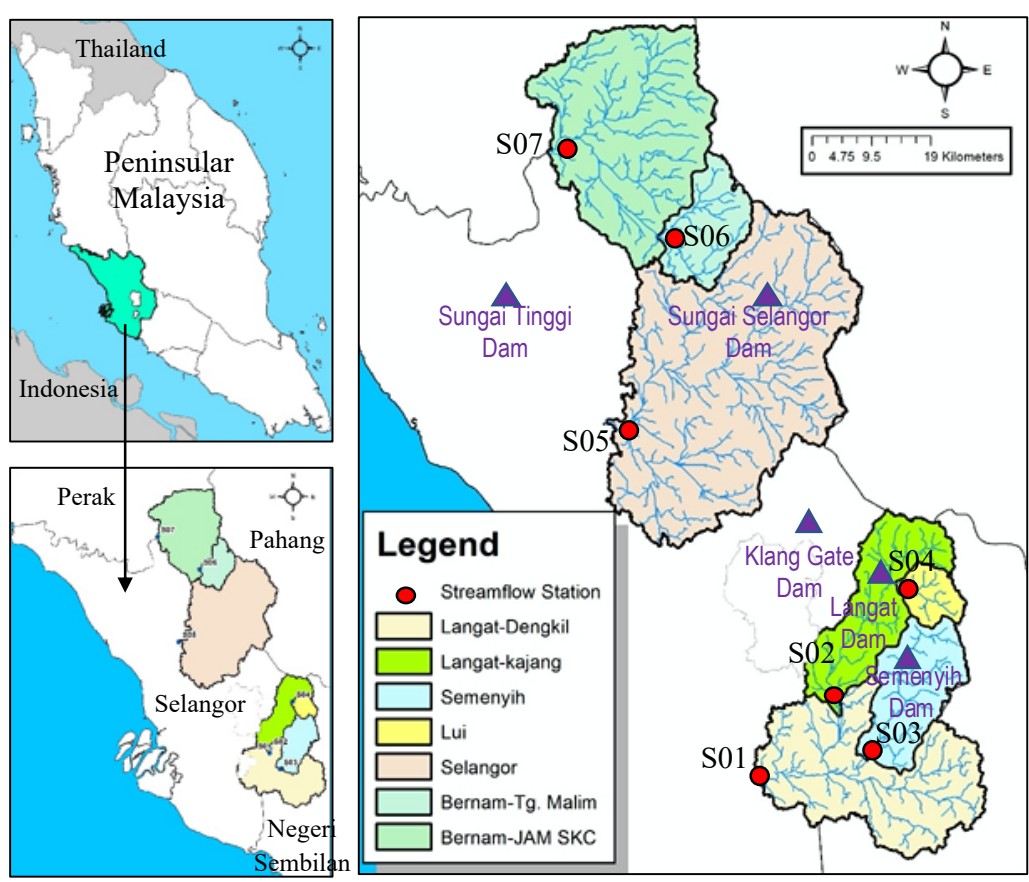


**Figure 1: River basin and streamflow station in Selangor.**

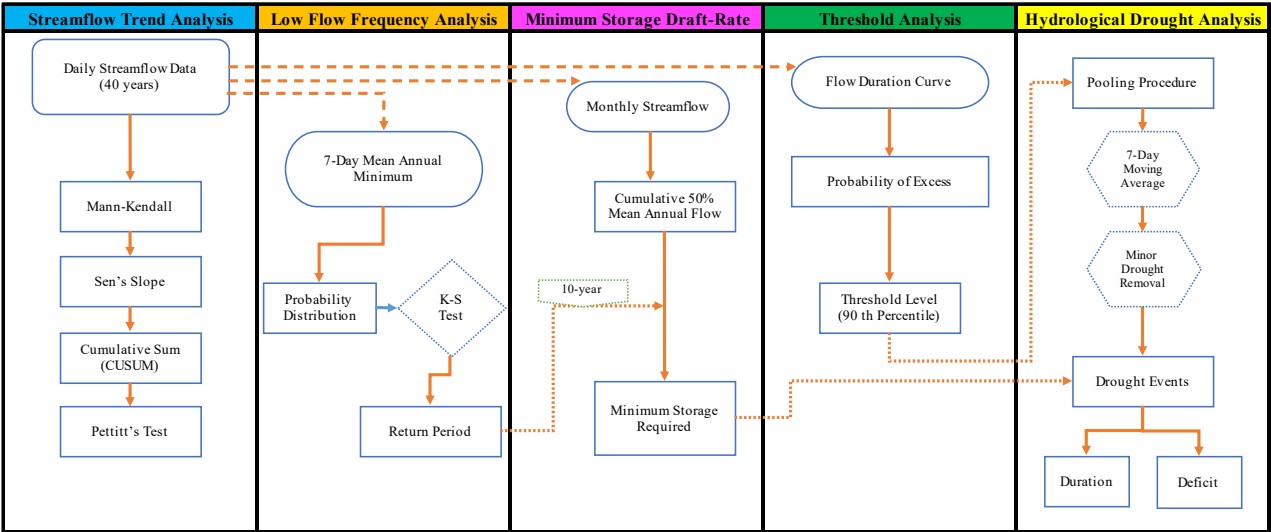

**Figure 2: Summary of methodology framework.**


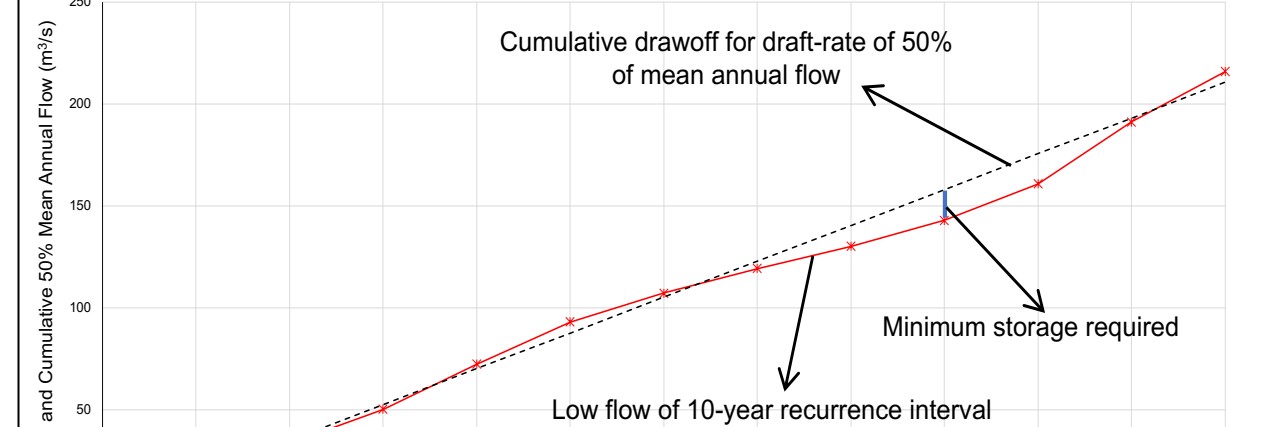

**Figure 3. Minimum storage required using mass curve analysis**

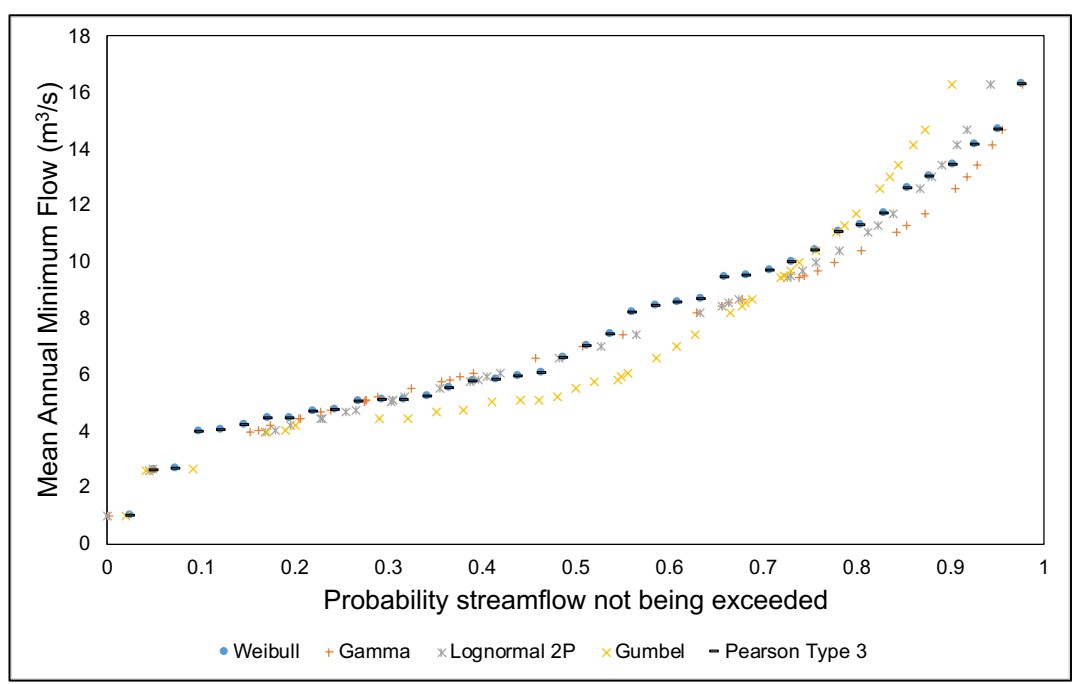

**Figure 4: Probability of mean annual minimum flow for station 1.**

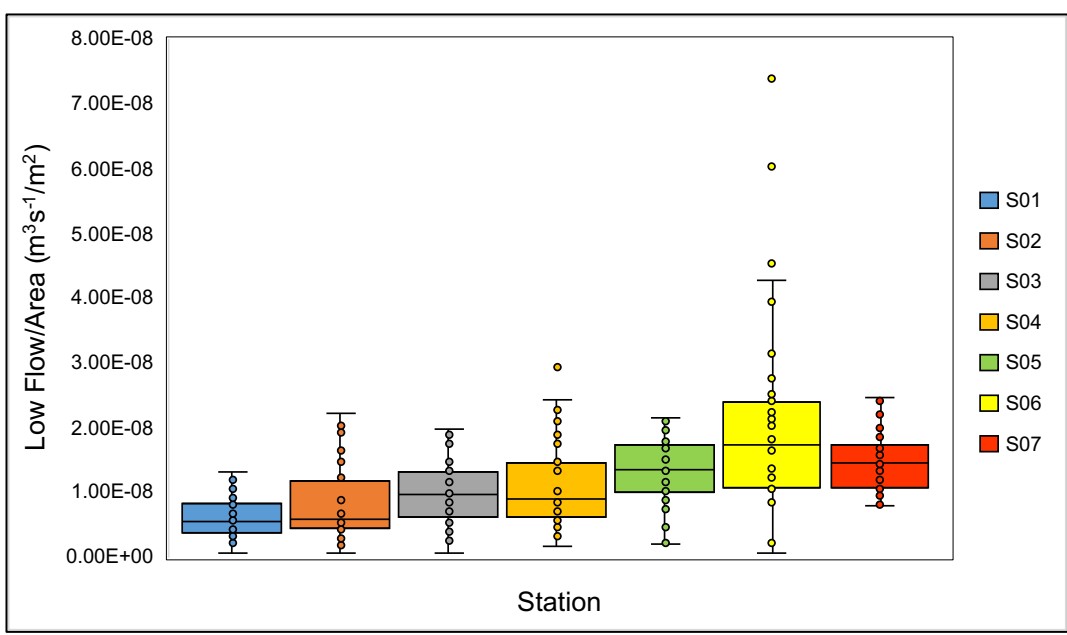

**Figure 5: The boxplot low flow per watershed catchment area.**

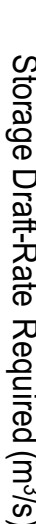

**Figure 6: Minimum storage draft rate with cumulative 50% mean flow (a) S01 (b) S02 (c) S03 (d) S04 (e) S05 (f) S06 (g) S07.**


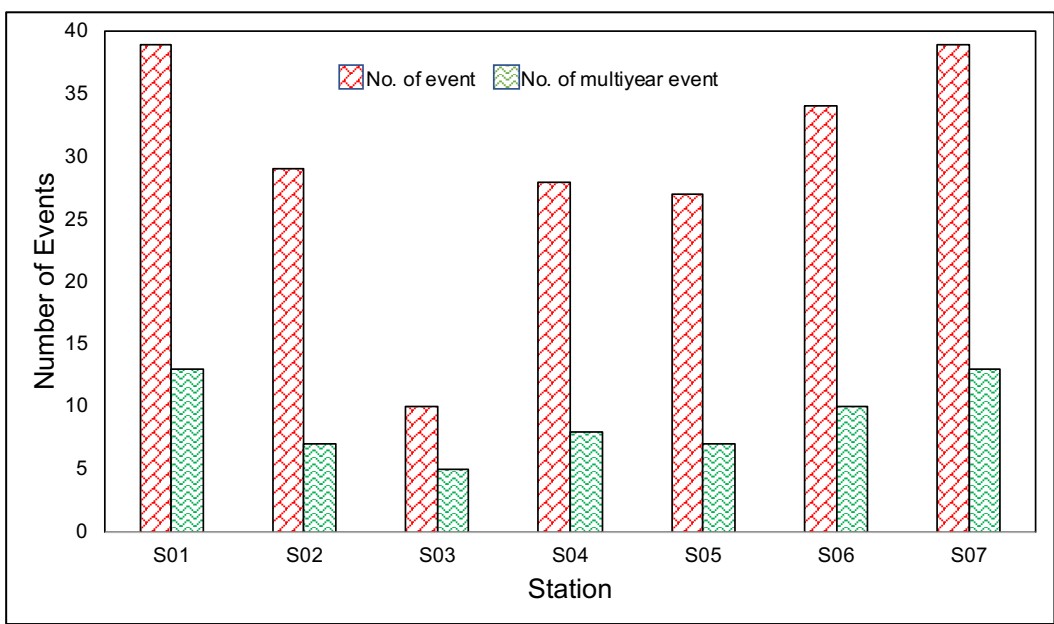

**Figure 7: Number of drought events.**

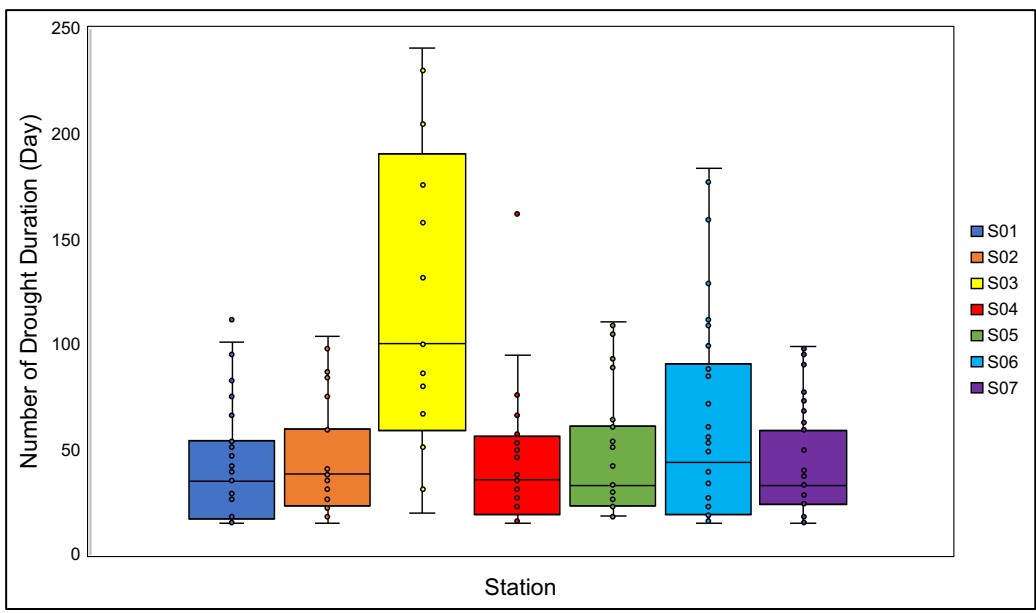

**Figure 8: The number of drought duration (days).**

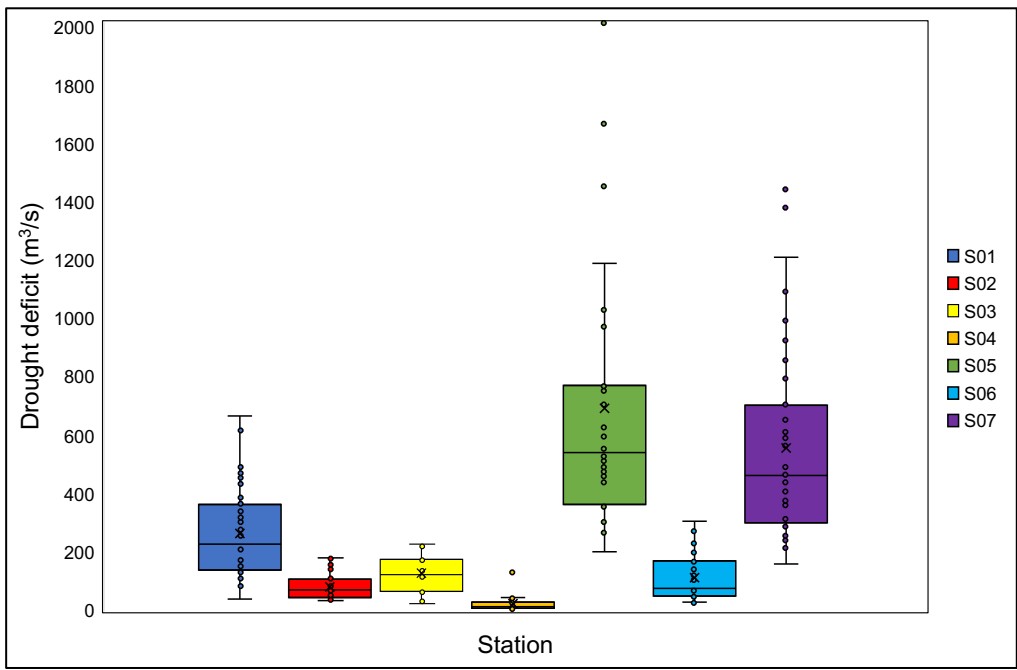


**Figure 9: The drought deficit for all station.**

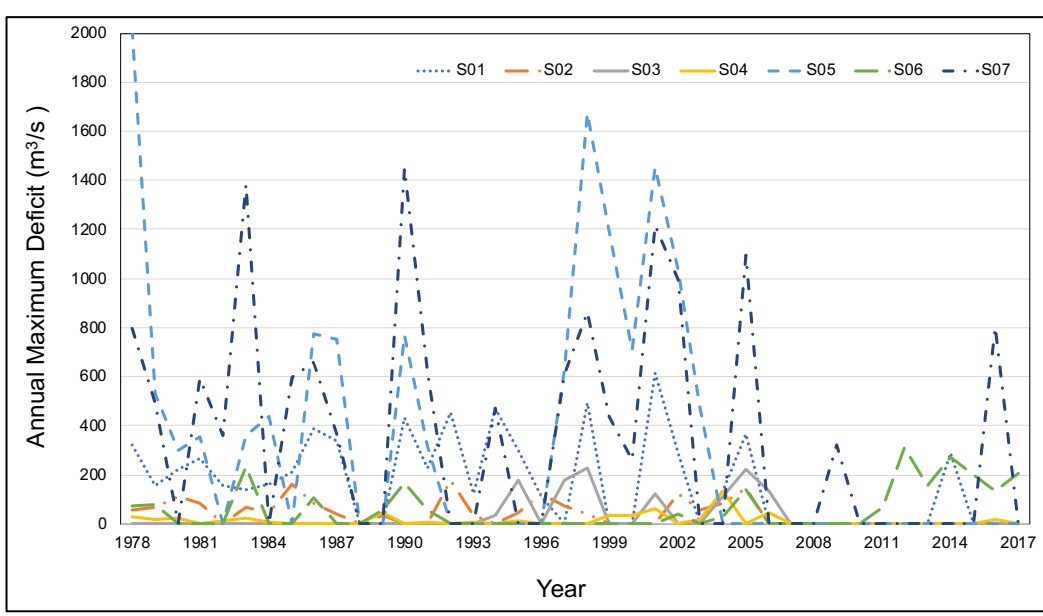

**Figure 10: Time series of annual maximum deficit (m³/s).**

**Table 1 The characteristics of streamflow gauging stations in Selangor.**

| Station No. | River Name | River Basin | Location Coordinate (WGS) | | Area (km²) | Affected by Reservoir |
|---|---|---|---|---|---|---|
| S01 | Langat-Dengkil | Langat | 02°51'20" N | 101°40'55" E | 1240 | No |
| S02 | Langat-Kajang | Langat | 02°59'40" N | 101°47'10" E | 380 | Yes |
| S03 | Semenyih | Langat | 02°54'55" N | 101°49'25" E | 225 | Yes |
| S04 | Lui | Langat | 03°10'25" N | 101°52'20" E | 68 | No |
| S05 | Selangor | Selangor | 03°24'10" N | 101°26'35" E | 1450 | Yes |
| S06 | Bernam- Tg. Malim | Bernam | 03°40'45" N | 101°31'20" E | 186 | No |
| S07 | Bernam-JAM SKC | Bernam | 03°48'15" N | 101°21'50" E | 1090 | No |

**Table 2 Probability density function for Gamma, Gumbel, Lognormal 2P and Pearson type-3 distributions**

| No. | Distribution | Probability Density Function | References |
|---|---|---|---|
| 1 | Gamma | $$f(x) = \frac{\beta^{-\alpha} x^{\alpha-1}}{\Gamma(\alpha)} exp\left(\frac{-x}{\beta}\right)$$ $\alpha > 0, \beta > 0, x > 0$, where $\alpha$ is the location parameter, and $\beta$ is the scale parameter | (Baran-Gurgul, 2018) |
| 2 | Gumbel | $$Fx(x) = exp\left[exp\left(\frac{x - \beta}{\alpha}\right)\right]$$ $-\infty < x < \infty$; $-\infty < \beta < \infty$; $\alpha > 0$. The $\alpha$ and $\beta$ parameters are parameters of scale and location. | (Zou et al., 2018) |
| 3 | Lognormal 2P | $$fx(x) = \frac{1}{\sqrt[x]{2\pi\beta^2}} e^{-\frac{(\ln x - \alpha)^2}{2\beta^2}}$$ $x > 0, \alpha > 0, \beta > 0$. | (Win and Win, 2014) |
| 4 | Pearson type-3 (PE3) | $$fx(x) = \frac{\lambda^\beta (x - \varepsilon)^{\beta-1} e^{-\lambda(x-\varepsilon)}}{\Gamma(\beta)}$$ $x \geq \varepsilon$. | (Bhatti et al., 2019) |

**Table 3 The statistical analysis for time series of streamflow (1978 - 2017).**

| Station No. | Mean Flow (m³/s) | Minimum Flow (m³/s) | Maximum Flow (m³/s) | Standard Deviation | Skewness | Kurtosis |
|---|---|---|---|---|---|---|
| S01 | 34.32 | 1.00 | 552.62 | 31.326 | 4.027 | 35.819 |
| S02 | 10.23 | 0.30 | 153.87 | 9.595 | 4.197 | 32.222 |
| S03 | 5.17 | 0.15 | 32.41 | 3.730 | 2.296 | 8.996 |
| S04 | 2.07 | 0.12 | 11.93 | 1.426 | 1.967 | 5.726 |
| S05 | 55.12 | 3.17 | 272.59 | 35.083 | 1.558 | 3.163 |
| S06 | 8.86 | 0.14 | 52.51 | 5.851 | 1.491 | 3.716 |
| S07 | 47.57 | 8.57 | 244.75 | 28.845 | 1.427 | 2.744 |

**Table 4 Trend analysis for time series period.**

| Station | Record Length | Mann-Kendall | Sen's Slope | Relative Change Within the Record (%) | Maximum Cumulative Sum (CUSUM) | Change Point (Year) | Value of $p$ (Pettitt's test) |
|---|---|---|---|---|---|---|---|
| S01 | 1978 - 2017 | 0.03 | 0.30 | 36.51 | 6 | 1996 | 0.1215 |
| S02 | 1978 - 2017 | 0.00 | 0.15 | 21.80 | 14 | 1997 | 0.0004 |
| S03 | 1978 - 2017 | -0.46 | -0.02 | -20.00 | 8 | 2006 | 0.1295 |
| S04 | 1978 - 2017 | 0.03 | 0.02 | 43.47 | 8 | 2007 | 0.0845 |
| S05 | 1978 - 2017 | 0.62 | 0.06 | 12.05 | 4 | 2005 | 0.4469 |
| S06 | 1978 - 2017 | -0.35 | -0.06 | -55.56 | 8 | 2009 | 0.0086 |
| S07 | 1978 - 2017 | 0.14 | 0.20 | 39.22 | 8 | 2005 | 0.2286 |

Note: For Mann-Kendall and Sen's slope, the positive values mean the increasing trends and the negative ones mean the decreasing trends

 **Table 5 Estimated parameters for the Gamma, Gumbel, Lognormal 2P and Pearson type 3 distributions.**

| Distribution | Parameters | | | | | | |
|---|---|---|---|---|---|---|---|
| | S01 | S02 | S03 | S04 | S05 | S06 | S07 |
| Gamma | $\alpha = 4.24$ | $\alpha = 1.92$ | $\alpha = 4.08$ | $\alpha = 3.20$ | $\alpha = 8.13$ | $\alpha = 1.83$ | $\alpha = 9.69$ |
| | $\beta = 1.78$ | $\beta = 1.53$ | $\beta = 0.55$ | $\beta = 0.24$ | $\beta = 2.52$ | $\beta = 2.10$ | $\beta = 1.60$ |
| Gumbel | $\sigma = 5.92$ | $\sigma = 1.92$ | $\sigma = 1.78$ | $\sigma = 0.57$ | $\sigma = 17.17$ | $\sigma = 2.55$ | $\sigma = 13.42$ |
| | $\mu = 2.89$ | $\mu = 1.64$ | $\mu = 0.87$ | $\mu = 0.33$ | $\mu = 5.94$ | $\mu = 1.68$ | $\mu = 5.47$ |
| Lognormal 2P | $\sigma = 8.09$ | $\sigma = 3.10$ | $\sigma = 2.45$ | $\sigma = 0.75$ | $\sigma = 20.65$ | $\sigma = 3.70$ | $\sigma = 16.46$ |
| | $\mu = 4.81$ | $\mu = 2.21$ | $\mu = 1.63$ | $\mu = 0.42$ | $\mu = 7.49$ | $\mu = 2.79$ | $\mu = 6.92$ |
| Pearson type 3 | $\alpha = 1.07$ | $\alpha = 2.46$ | $\alpha = 2.87$ | $\alpha = 7.78$ | $\alpha = 0.60$ | $\alpha = 2.00$ | $\alpha = 0.63$ |
| | $\beta = 5.00$ | $\beta = 5.00$ | $\beta = 5.00$ | $\beta = 5.00$ | $\beta = 5.00$ | $\beta = 5.00$ | $\beta = 5.00$ |

**Table 6 The values of the Kolmogorov-Smirnov (KS) test**

| Station | Distribution | KS test statistics | *P*-Value | Rank |
|---------|--------------|--------------------|-----------|------|
| S01 | Gamma | 0.09 | 0.9110 | 2 |
| | Gumbel | 0.09 | 0.8581 | 3 |
| | Lognormal 2P | 0.08 | 0.9626 | 1 |
| | Pearson type 3 | 0.23 | 0.0204 | 4 |
| S02 | Gamma | 0.09 | 0.9074 | 2 |
| | Gumbel | 0.10 | 0.8241 | 4 |
| | Lognormal 2P | 0.09 | 0.8823 | 3 |
| | Pearson type 3 | 0.07 | 0.9796 | 1 |
| S03 | Gamma | 0.09 | 0.8810 | 2 |
| | Gumbel | 0.09 | 0.8984 | 1 |
| | Lognormal 2P | 0.10 | 0.8275 | 3 |
| | Pearson type 3 | 0.12 | 0.5866 | 4 |
| S04 | Gamma | 0.10 | 0.8181 | 2 |
| | Gumbel | 0.11 | 0.7430 | 3 |
| | Lognormal 2P | 0.09 | 0.9004 | 1 |
| | Pearson type 3 | 0.19 | 0.0989 | 4 |
| S05 | Gamma | 0.08 | 0.9401 | 1 |
| | Gumbel | 0.09 | 0.8956 | 3 |
| | Lognormal 2P | 0.09 | 0.9062 | 2 |
| | Pearson type 3 | 0.35 | 0.0001 | 4 |
| S06 | Gamma | 0.12 | 0.6354 | 4 |
| | Gumbel | 0.07 | 0.9905 | 1 |
| | Lognormal 2P | 0.10 | 0.8296 | 2 |
| | Pearson type 3 | 0.11 | 0.7418 | 3 |
| S07 | Gamma | 0.10 | 0.8406 | 3 |
| | Gumbel | 0.09 | 0.8990 | 2 |
| | Lognormal 2P | 0.08 | 0.9608 | 1 |
| | Pearson type 3 | 0.36 | 0.0001 | 4 |


**Table 7 The return period of low flow at all streamflow stations.**

| Station No. | Low Flow at Return Period (m³/s) | | | | | | |
|---|---|---|---|---|---|---|---|
| | 1-year | 2.3-year | 5-year | 10-year | 25-year | 50-year | 100-year |
| S01 | 21.42 | 18.19 | 15.27 | 12.63 | 9.13 | 6.49 | 3.85 |
| S02 | 10.60 | 8.83 | 7.24 | 5.80 | 3.89 | 2.44 | 1.00 |
| S03 | 6.44 | 5.45 | 4.55 | 3.73 | 2.66 | 1.84 | 1.02 |
| S04 | 2.25 | 1.90 | 1.58 | 1.29 | 0.91 | 0.62 | 0.34 |
| S05 | 48.40 | 41.54 | 35.35 | 29.72 | 22.29 | 16.67 | 11.05 |
| S06 | 13.09 | 10.91 | 8.93 | 7.14 | 4.78 | 2.98 | 1.19 |
| S07 | 34.56 | 30.14 | 26.15 | 22.53 | 17.74 | 14.12 | 10.49 |

Note: 10-year low flow return period will be used in the determination of minimum storage draft-rate.
