# Peer review of "Assessment of probability distributions and minimum storage draftrate analysis in the equatorial region"

_Natural Hazards and Earth System Sciences, 2020_

## Referee Comment (RC1) · Anonymous Referee #1 · 9 May 2020

**Review for manuscript "Assessment of probability distributions and minimum storage draft-rate in the equatorial region"**

**Authors:** Hasrul Hazman Hasan, Siti Fatin Mohd Razali, Nur Shazwani Muhammad, Zawawi Samba Mohamed, Firdaus Mohamad Hamzah

**Journal:** Natural Hazards and Earth System Sciences

**Summary**

The study by Hasan et al. focuses on low flows, drought, and minimum storage draft-rates in seven catchments in the Selangor region in Malaysia. The study consists of four types of analyses: (1) a non-parametric trend analysis on annual mean, minimum, and maximum flows using the Mann-Kendall and Sen's slope tests; (2) a low flow frequency analysis on annual minimum flow using the Lognormal 2P distribution; (3) an analysis of drought characteristics determined using a fixed drought threshold at the 90th flow percentile; and (4) the determination of minimum storage draft rates necessary to ensure sufficient water supply during low flow periods.

**General remarks**

The study performs a variety of analyses related to low flows and drought and in my opinion has several deficiencies. (1) It does not seem to follow a clear aim and motivation and lacks the specification of a research question; (2) it has an unclear structure and shows elements belonging to Introduction, Methods, Results, Discussion, Conclusions all over the place (i.e. not all introductory material is in the introduction,…); (3) the method descriptions are confusing and it is hard to tell how the analysis was exactly done. I was only able to understand what was approximately done when I finished reading the conclusions; (4) the presentation of the results could be significantly improved; (5) a novel aspect is missing, which leads to insignificant conclusions. I do not think that this study is publishable in NHESS.

I still discuss some major points below which may help to improve the study design and presentation.

**Major points**

- **Title:** I would replace 'in the equatorial region' by 'in Malaysia'.
- **Abstract:** The abstract is missing a clear problem statement. The study region of interest should be mentioned. I would give it a clear structure by listing the four elements of the analysis: (1) trend analysis, (2) low flow frequency analysis, (3) drought analysis, and (4) storage draft rate analysis. The abstract should also include a short summary of the main findings and end with a concluding statement (this requires a clear problem statement at the beginning).
- **Introduction:** The introduction needs a clear research question and should introduce the problem and some background knowledge related to this research question (or questions). Currently, the introduction lists various statements related to low flows and droughts but does not tell a compelling story. The introduction would profit from a clear distinction between low flows, droughts, and water scarcity (for a discussion on these different concepts see e.g. [*Van Loon et al.*, 2016]). In addition, a short introduction to the concept of 'storage rate' should be provided (e.g. does storage refer to reservoir storage or another type of

storage?). I suggest to restructure the introduction as follows: (1) introduce why are droughts, low flows, and water scarcity important and what is the relationship between the three, (2) introduce factors influencing drought and water scarcity characteristics, (3) introduce the storage-draft rate concept and how this is related to drought, (4) provide a short introduction of study area and the problem you are trying to solve, (5) state research question, and (6) provide a short overview of methods used to answer this question.

- **Data:** The study lacks a proper introduction of the dataset used for the analysis. The following specifications are necessary: are you working with observed or simulated streamflow data? Are the streamflow time series natural or influenced by water abstraction and storage (at least some of them seem to be influenced)? Why are inconsistencies a problem? What types of streamflow regimes do the catchments represent (i.e. what is the seasonality of the Indian and Asian monsoons)?

- **Methodology:** In my understanding, the analysis consists of four main steps: (1) Trend analysis of annual mean, maximum, and minimum flows, (2) low flow frequency analysis based on annual minimum flows, (3) analysis of drought characteristics for individual events, and (4) storage draft analysis. Is this correct. If this is what was actually done, I would restructure the methods section accordingly. It is unclear which types of variables are used for which type of analysis. I only figured out e.g. which variables were of interest in the trend analysis when I started to look at the tables presented in the Results section. The methods descriptions are confusing and unclear and include a lot of unnecessary detail instead of providing essential information. I do for example not understand why a detailed description of Flow Duration Curves is necessary (these were just used to determine the drought threshold, right?). Or what does the description of plotting positions do in the methods section (I did not find any results relating to plotting positions)? In my opinion, the detailed description of the Mann-Kendall test can be removed and be replaced by an appropriate reference (l. 131-157). Instead, it should be specified (a) which distributions were used to fit the low flow datasets and why (i.e. which distributional properties are important here), (b) how low flow is defined (based on the results I believe as the minimum annual flow but this is not clear from the methods section), (c) for which variable/events return periods were determined, (d) whether the determination of return periods relies on empirical or theoretical distributions, (e) which drought characteristics were analyzed in the below threshold drought analysis, (f) whether a short pooling window of 7 days (l.220) actually guaranteed independence of events, (g) whether minor droughts were removed or not (methods section says yes, results section says no (l. 297)), (h) what the storage-draft rate method does and what kind of storage it refers to (an illustration of the concept would help).

- **Results:** The results section contains several paragraphs actually belonging to the methods and introduction sections (e.g. l. 246-250: and by the way I thought the trend analysis was performed using the non-parametric Mann-Kendall test and not linear regression). There is even a statement that belongs to the introduction describing the 'primary purpose' of this study (l. 260-261). I would restructure according to the restructuring also suggested for the Methods section: (1) Results of trend analysis, (2) results of low flow frequency analysis, (3) results of drought characteristics analysis, and (4) results of storage rate analysis. And also here, it always needs to be clear which variables the results refer to. I would in some instances replace results presented in tables by figures. This particularly concerns l.300 -351. I would try to visualize these results instead of presenting them as plain text. E.g. number of

events as barplots, durations, and deficits as boxplots for all stations. This would allow for a comparison across stations. In addition, you could also plot deficit time series per station to compare particular events.

- **Discussion:** The discussion presents a lot of material that in my opinion belongs to the introduction (l. 393-411). I would instead discuss the implications of your findings for water management in the region.

- **Conclusions:** Instead of providing a summary of the methods, focus on the insights we gain from this study. Currently this seems to be: 'Based on the analysis of the study, the estimated minimum storage-draft rates for each station cannot meet the water demand during low flow at specific return periods, which is 10-year recurrence interval for this research.' (l. 448-449). Formulating conclusions will be easier once you have identified a clear research question.

- **References:** Should be carefully checked. There is at least one duplicate (Sarailidis et al. 2019), and I would consistently use lower caps for nouns (e.g. Bakanogullari et al. 2014).

- **Language:** The article needs editing with respect to the use of tense and sentence structure. Some redundant information can be removed (e.g. l. 102 and l. 107).

**Figures and Tables:**

- **Figure 1:** I would indicate the locations of the dams mentioned in l.90-99 if they are important for the analysis. But I am still unsure whether the storage-rate refers to reservoir storage or something else. I would reduce the density of the stream network displayed in order to increase the distinctiveness of the colors.

- **Figure 2:** Is this figure really needed?

- **Figure 3:** Indicate that outliers are not displayed?

- **Figure 4:** Increase legend font, provide one legend for all subplots not per subplot. What does the dark grey bar mean? Increase size of axis labels.

- **Table 1:** Can in my opinion be removed as information is also contained in Figure 1.

- **Table 2:** Introduce in methods section, reference should be provided for each distribution.

- **Table 4:** It seems as if trends were not only determined over the whole period but also for very short time periods of 7 years. This sub period analysis does in my opinion not make sense. I think I would plot time series of mean, minimum, and maximum flow for each catchment to illustrate the trends and instead remove the sub period analysis.

- **Table 5:** I find it strange that no 3-parameter distributions were tested as extreme values are usually insufficiently represented by 2-parameter distributions.

- **Table 6:** The p-values should lie in the range of [0,1]. Were the column names mixed up? I would indicate for which distributions and catchments, H0 of 'the distribution of the sample corresponds to the theoretical distribution' was rejected.

- **Table 8:** can in my opinion be removed as you just focused on a threshold of Q90. By the way, I would talk about Q10, to consistently refer to non-exceedance probabilities throughout the paper.

- **Table 10:** Is this table related to Figure 4, and if so how or could it even be removed?

**Minor points**

Trend detection and attribution is a pretty active research area and I would not agree that we are 'beginning to pay more attention to trend analysis' (l. 118).

A goodness-of-fit test rejects or non-rejects a hypothesis but does not 'accept a fit' (l. 163).

The return period in a univariate setting is defined as T=1/(1-p), where p is the non-exceedance probability, i.e. T it is not the probability of occurrence itself (l. 188).

l.353-359: move material to introduction.

l. 363-367: move to methods section

No further editing suggestions are provided as the manuscript in my opinion needs to be completely revisited.

**References used in this review**

Van Loon, A. F. et al. (2016), Drought in a human-modified world: Reframing drought definitions, understanding, and analysis approaches, *Hydrol. Earth Syst. Sci.*, *20*(9), 3631–3650, doi:10.5194/hess-20-3631-2016.

---

## Referee Comment (RC2) · Anonymous Referee #2 · 20 May 2020

The manuscript focuses on analyzing streamflow in seven station in the Selangor state (Malaysia). The paper is interesting and presents an acceptable analysis, however, in my opinion there are a few drawbacks in the paper, which can be eliminated by carrying out some major revisions following the list of comments below.

MAJOR PROBLEMS:

My main concern refers to the trend analysis. First, why the trend analysis has been performed on 5 8-years sub-periods? Then, are the trends statically significant? Recommendation: specify if the trends are statically significant and the confidence level considered. Finally, the authors applied both the MK and the Sen's Slope to evaluate

the trend sign, but besides the trend sign it could be interesting to detect the trend magnitude. I suggest to apply the Sen's Slope for the evaluation of the slopes of the trends and the Mann–Kendall test for the assessment of the statistical significance.

Which is the influence of the dams on the results of this study?

Can the authors better explain the aims of the paper?

Finally, the author simply describe the results present in the study, and not discuss those results in depth. The authors should try to improve the discussion to underline the added value of their work compared to other similar in the same area and in different areas of the world.

MINOR COMMENTS:

The English grammar, syntax and punctuation should be improved and I recommend professional proofreading by a native speaker.

Add some references for the Mann-Kendall test, for the Sen's slope estimator and for each distribution of Table 2.

Some references in the text are missing in the references list: Kannan et al. (2018); Sarailidis et al. (2019b). In the references list the latter is a duplicate of Sarailidis et al. (2019a)

A reference in the references list is in the wrong position: Van Loon and Van Lanen (2013)

Figure 1: I think that this figure is not sufficiently informative and it must be greatly improved. Can the authors try to better identify the different sub-basins? Moreover, can the authors show the position of the dams? Finally, it is important to show the localization of the basins within a larger area and to add the coordinates because in this form this figure is hard to understand for a non-Malaysian reader.

Figure 3: please describe what the boxes and the whiskers mean. Which percentiles

[Figure]

or interquartile ranges are represented?

---

## Author Response (AR1)

**Editor comment on "Assessment of probability distributions and minimum storage draft-rate analysis in the equatorial region" by Hasrul Hazman Hasan et al.**

**Editor comment:**
Dear Authors,
Based on the reviewers' comments and your replies, I believe that your manuscript requires major revisions. When revising your manuscript according to the valuable suggestions and comments by the referees, I expect that you will address the following key issues.

**Authors Response:**
We want to thank you for your constructive comments. We have improved the whole manuscript based on your suggestions.

**Editor comment:**
To highlight the difference between drought and low flow in the introduction (a good reference could be also found in Chapter 8 of the following book: Jose D. Salas, Charles N. Kroll, Antonino Cancelliere, Bonifacio Fernández, Jose A. Raynal, and Dong R. Lee Statistical Analysis of Hydrologic Variables: Methods and Applications. 2019).

**Authors Response:**
Thank you for your recommendation. We have revised the introduction part based on your recommendation in lines 66-76, page 3, as follows:

A hydrological drought is a natural event with streamflow deficits in duration and volume (Kubiak-Wójcicka and Bąk, 2018). It is believed that not every low flow event can be considered a hydrological drought, and that one hydrological drought can consist of several low flows (Teegavarapu et al., 2019). It is not advisable to equate hydrological drought with low flow or other related hazards. Low flow is a term that is often used, referring to low flow discharge. Low flow is often defined by minimum annual series which do not reflect a hydrological drought in all years. Fleig et al. (2006) were distinguish between hydrological droughts and low flow characteristics. For some specific purpose, the main feature of drought is said to be the water deficit. Low flows are usually observed during a drought, but they only feature one aspect of the drought, namely the magnitude of drought. Low flow analysis is described as analyses that attempt to understand the short-term physical development of flows at a point along a river. The minimal annual $n$-day average discharge is the most widely used low flow index.

**Editor comment:**
To clarify both the objectives and novelties of the paper with respect to the state of the art knowledge.

**Authors Response:**
Thank you for your comment.
The primary purpose of this study are: (1) to arbitrate the trend analysis of streamflow for 40 years; (2) to determine the best-fitted distribution of probability for each station for low flow frequency analysis; (3) to evaluate the hydrological drought characteristics, including severity, duration and magnitude; (4) to determine the minimum storage draft rates of seven catchments

in Selangor region in Malaysia. This study is essential to understand the concept of low flow, drought characteristics, and the predictive significance of river storage-draft rates in managing sustainable water catchment. The results are useful for developing measures to maintain flow variability and can be used to develop policies for risk management.

This paper discussed the advantage of hydrological drought analysis with two variables (deficit and duration) over more common single value analysis, e.g. annual minimal discharge. The threshold level method was applied for the period 1978-2017 at seven stations in the Selangor basin, which represents the most significant sample used in Selangor, referring to low flow analysis. For the definition of the droughts value, Q90 was selected as the threshold level, since the objective of the study was the spatial and temporal characteristics of extreme (extensive) droughts in the Selangor basin. The study of the relationship between deficits and the duration of droughts and the physical-geographic characteristics of the basin is not the aim of this paper but deserved further interest in further research.

These relationships may lead to a significant determination of the mean deficit or duration, which is vital for further regional statistical analysis, such as estimating the deficit on ungauged basins with different return periods. Hydrological drought is a complex issue in terms of its potential causes and ecosystem and societal impacts. Hence, it is crucial to understand the mechanisms of its beginning, development and termination. Establishing a robust quantification of hydrological droughts is essential, — for example, the findings should have more applicability in water management. A further stage is to bridge the gap between science, management and policymakers in order to apply accumulated research expertise in the area. This paper will help the professional public better understand the hydrological problems Drought and time series of drought deficits and durations derived in this paper are useful in understanding the equatorial region's drought characteristics.

 **Editor comment:**
To provide a clear logical connection among the methodological approaches adopted in the study, since so far they appear barely linked.

**Authors Response:**
Thank you for your comment.
The study consists of four types of analyses: (1) a non-parametric trend analysis on annual mean, minimum, and maximum flows using the Mann-Kendall and Sen's slope tests; (2) a low flow frequency analysis on annual minimum flow using the Lognormal 2P distribution; (3) an analysis of drought characteristics determined using a fixed drought threshold at the 90th flow percentile; and (4) the determination of minimum storage draft rates necessary to ensure sufficient water supply during low flow periods.

In developing water resources management for the Selangor basin, Malaysia, the numerous streamflow records were analysed to determine their flow characteristics. The objective was to present the results in a format which is well adapted for use by water resource planners in preliminary designs to obtain optimum benefits from the water resources. The first analysis is non-parametric trend analysis on annual mean, annual minimum, and annual maximum flow using the Mann-Kendall and Sen's slope tests. Next, a low flow frequency analysis on annual minimum flow and analysis of drought characteristics determined using a fixed drought threshold at the 90th flow percentile. The primary analysis is the determination of necessary minimum storage draft rates to ensure sufficient water supply during low flow periods. The

minimum storage draft rate is the probable amounts of storage required to meet specified levels of sustained demand for water. The method used gives not only information concerning amounts of storage, but also information concerning the probable amounts of excessive flows that might be used for this storage, both of which are the functions of carryover period and the recurrence interval where hereafter are referred as frequency mass curve analysis. The analysis also determines the probable variation of the streamflow for each month of the year. These probable variations, or frequency analyses, are obtained by fitting the data to any or all of the following four distribution functions: Gamma, Gumbel, Lognormal 2P and Pearson type 3.

In the frequency of occurrence estimation, the series of annual maximum storages can be assumed to be independent and ranked in an ascending or descending order with a plotting position assigned to each value, according to the rank and sample size. In order to estimate storages beyond the range of probabilities given by the assigned plotting positions, it is necessary to assume a form of a distribution function. Here it is assumed that the series followed an extreme value type 1 (EV1) distribution. From the fitted distributions, deficits expressed in cumulative of low flow for return periods of 10 years and yield levels 50% of the mean flow. A negative quantity denotes the quantity available for storage. By visualising straight lines with a slope equal to the rates of constant demand superimposed on the mass curve frequency, the amounts of needed storage for any period of carryover at any of the specified levels of probability can be obtained as the difference between the straight line ordinates and that of the frequency mass curve (Figure 1).

[Figure]

Figure 1. Minimum storage required using mass curve analysis

**Editor comment:**
To justify why operation rules are not taken into account in the reservoir storage analysis.

**Authors Response:**
Thank you for your comment.
In this paper, the calculated necessary storage cannot be interpreted as actual necessary storage for reservoir design and water management, since the actual status of channel capacity and water withdrawal at a given location are not accounted for in the calculation. It is only a sign of discharge variability in the unit of necessary storage to maintain certain constant flow levels

for water use in downstream areas. Ascertain flow levels, the mean annual flow or some percentage of it is used to help users to imagine the assumptions easily. This research can answer the research question which is to suppose a drought manager considers how much water is necessary to be stored before the dry season starts and to keep supplying water equal to the long-term mean of river discharge, which is indicated by mass curve analysis.

In this paper, necessary storage is proposed as a signature of hydrological variability in time. Its advantage over other statistical indicators is evident as it is in human terms and has a direct implication for the ease and difficulty of water resource management. This study is primarily staged before any developments or withdraw freshwater from river take place. The reservoir storage analysis is more detail and different within different water management. An iterative procedure needs to done to determine the maximum storage in the reservoir to meet demand plus evaporative losses. This study was focusing on streamflow data analysis only without consideration of evaporative losses. Quantification of the storage effect of a dam reservoir is always required for evaluating the dam in the entire river basin. However, as the reservoir storage–outflow relationship is dependent on both the morphological characteristics of the reservoir and the dam operation rule, it generally becomes nonlinear. Differently from the linear reservoir case, it is not easy to quantify the storage effect or to define the storage coefficient.

**Editor comment:**
To have your manuscript proofread by a native English speaker.

**Authors Response:**
Thank you for your comments. We have submitted the manuscript for the professional proofreading. We have attached the proofreading certificate in the attachment.

**References**

Fleig, A. K., Tallaksen, L. M., Hisdal, H. and Demuth, S.: A global evaluation of streamflow drought characteristics, Hydrol. Earth Syst. Sci., 10(4), 535–552, DOI:10.5194/hess-10-535-2006, 2006.

Kubiak-Wójcicka, K. and Bąk, B.: Monitoring of meteorological and hydrological droughts in the Vistula basin (Poland), Environ. Monit. Assess., 190(11), 87–100, DOI:10.1007/s10661-018-7058-8, 2018.

Teegavarapu, R. S. V, Salas, J. D. and Stedinger, J. R.: Statistical Analysis of Hydrologic Variables., 2019.

**Attachment**

[Figure]

PSUK Communications LTD
83 Ducie Street
Manchester, UK
M1 2JQ

Email: info@qualityproofreading.co.uk
Tel: (+44) 0330 822 0143
VAT: GB 227 3104 40
Company number: 09305305

**Certificate for Proofreading and Editing Services Undertaken**

16th April 2020

Authors:

Hasrul Hazman Hasan, Siti Fatin Mohd Razali, Nur Shazwani Muhammad, Zawawi Samba Mohamed, Fird

This document certifies that the manuscript titled "Assessment of probability distributions and minimum storage draft-rate analysis in the equatorial region" was proofread and edited by Proofreading Service UK.

The editor aimed to ensure that the author's intended meaning was not altered during the review.

All of the suggested amendments were tracked with the Microsoft Word "Track Changes" feature. Therefore, the author had the option to reject or accept each change individually.

Kind regards,

Proofreading Service UK
Summary - The study by Hasan et al. focuses on low flows, drought, and minimum storage draft-rates in seven catchments in the Selangor region in Malaysia. The study consists of four types of analyses: (1) a non-parametric trend analysis on annual mean, minimum, and maximum flows using the Mann-Kendall and Sen's slope tests; (2) a low flow frequency analysis on annual minimum flow using the Lognormal 2P distribution; (3) an analysis of drought characteristics determined using a fixed drought threshold at the 90th flow percentile; and (4) the determination of minimum storage draft rates necessary to ensure sufficient water supply during low flow periods.

General remarks - The study performs a variety of analyses related to low flows and drought and in my opinion has several deficiencies. (1) It does not seem to follow a clear aim and motivation and lacks the specification of a research question; (2) it has an unclear structure and shows elements belonging to Introduction, Methods, Results, Discussion, Conclusions all over the place (i.e. not all introductory material is in the introduction,…); (3) the method descriptions are confusing and it is hard to tell how the analysis was exactly done. I was only able to understand what was approximately done when I finished reading the conclusions; (4) the presentation of the results could be significantly improved; (5) a novel aspect is missing, which leads to insignificant conclusions. I do not think that this study is publishable in NHESS. I still discuss some major points below which may help to improve the study design and presentation.

**Authors Response:**

We would like to thank you for your constructive comments. We have improved the whole manuscript based on your suggestions.

**Referee #1 Comment:**

Major points - Title: I would replace 'in the equatorial region' by 'in Malaysia'.

**Authors Response:**

Thank you for your recommendation. Malaysia is located in the equatorial region. We want to acquaint Malaysia as one of the countries located in the equatorial region, therefore would like to keep the current title.

**Referee #1 Comment:**

Abstract - The abstract is missing a clear problem statement. The study region of interest should be mentioned. I would give it a clear structure by listing the four elements of the analysis: (1) trend analysis, (2) low flow frequency analysis, (3) drought analysis, and (4) storage draft rate analysis. The abstract should also include a short summary of the main findings and end with a concluding statement (this requires a clear problem statement at the beginning).

**Authors Response:**

Thank you for your suggestion. We agree with the reviewer. We have revised the abstract based on your recommendation with a summary of the main findings and clear problem statements.

**Referee #1 Comment:**

Introduction - The introduction needs a clear research question and should introduce the problem and some background knowledge related to this research question (or questions). Currently, the introduction lists various statements related to low flows and droughts but does not tell a compelling story. The introduction would profit from a clear distinction between low flows, droughts, and water scarcity (for a discussion on these different concepts see e.g. [*Van Loon et al.*, 2016]). In addition, a short introduction to the concept of 'storage rate' should be provided (e.g. does storage refer to reservoir storage or another type of storage?). I suggest to restructure the introduction as follows: (1) introduce why are droughts, low flows, and water scarcity important and what is the relationship between the three, (2) introduce factors influencing drought and water scarcity characteristics, (3) introduce the storage-draft rate concept and how this is related to drought, (4) provide a short introduction of study area and the problem you are trying to solve, (5) state research question, and (6) provide a short overview of methods used to answer this question.

**Authors Response:**

Thank you for your recommendation. We have revised the introduction part based on your recommendation.

**Referee #1 Comment:**

Data - The study lacks a proper introduction of the dataset used for the analysis. The following specifications are necessary: are you working with observed or simulated streamflow data?

**Authors Response:**

The analysis is based on the observed streamflow data. Streamflow data were obtained from the Department of Irrigation and Drainage Malaysia, which covers approximately 40 years (1978 to 2017) of records for all streamflow gauging stations. Precautions were taken to ensure reasonable low flow regimes are captured. The daily observed streamflow data have consistent statistical properties and analysis of streamflow for determining the threshold level values to drought analysis. Lastly, the minimum storage draft rate required for Selangor was determined using a mass curve analysis.

**Referee #1 Comment:**

Are the streamflow time series natural or influenced by water abstraction and storage (at least some of them seem to be influenced)?

**Authors Response:**

Many factors influence the streamflow time series. The importance of natural hydrological regimes in maintaining the integrity of rivers has been widely recognised. Anthropogenic pressures, such as dams, point source discharges, surface water abstractions, and hydropower, may modify the natural regime of a river with a negative impact on water ecosystems.

**Referee #1 Comment:**

Why are inconsistencies a problem? What types of streamflow regimes do the catchments represent (i.e. what is the seasonality of the Indian and Asian monsoons)?

**Authors Response:**

We have explained the details in the study area part (line 161-168, pages 5-6). The equatorial climatic regions are influenced by two monsoons, which are the southwest Indian monsoon and the northeast Asian monsoon contribute two rainy seasons with a significant amount of storm events resulting in a mean annual rainfall of about 2500 mm (Mamun et al., 2010). Even though Selangor is located in the humid region, it occasionally encounters drought periods. Dry

spells, low rainfall, and increased soil impermeability due to population growth are the leading causes of low flow events. The low flow usually refers to a stream regime that indicates the average annual streamflow variability associated with the regional climate's annual cycle. A stream's regime can display one or more low flow events depending on the climate. Two rainy and two dry seasons represent the equatorial climate, and the two streamflow regimes have two corresponding periods of high flow and low flow.

**Referee #1 Comment:**
Methodology - In my understanding, the analysis consists of four main steps: (1) Trend analysis of annual mean, maximum, and minimum flows, (2) low flow frequency analysis based on annual minimum flows, (3) analysis of drought characteristics for individual events, and (4) storage draft analysis. Is this correct. If this is what was actually done, I would restructure the methods section accordingly. It is unclear which types of variables are used for which type of analysis. I only figured out e.g. which variables were of interest in the trend analysis when I started to look at the tables presented in the Results section. The methods descriptions are confusing and unclear and include a lot of unnecessary detail instead of providing essential information. I do for example not understand why a detailed description of Flow Duration Curves is necessary (these were just used to determine the drought threshold, right?).

**Authors Response:**
Thank you for your comments. We have explained clearly in the manuscript (line 249-251, page 9).
Flow Duration Curve (FDC) steps are essential because FDC can describe the ratio of a specified percentage of time with discharge is equal to or surpassed (Croker et al., 2003; Mohamoud, 2008; Vogel and Fennessey, 1994), which reflects the relationship between streamflow magnitude and length of time that relates to the average percentage of time a specific flow is exceeded (Sung and Chung, 2014). Thus, FDC consists of a complete record of streamflow magnitude for 40 years.

**Referee #1 Comment:**
Instead, it should be specified (a) which distributions were used to fit the low flow datasets and why (i.e. which distributional properties are essential here),

**Authors Response:**
Thank you for your recommendation. We have revised the manuscript in lines 341-352, page 12.

The primary aim of the probability distribution fitting is to represent the low flow probability most accurately. Among all the stations, it was found that among all distributions, the Lognormal 2P yielded the most cases of best-fit distributions, while the Gumbel and Gamma yielded the second and third amount of best-fits respectively. Comparatively, it is proposed that 2P Lognormal distributions predict low-flow discharges for all the rivers under analysis, which can be used in water quality and quantity management at gauged and ungauged areas. When the best fit probability distribution of the low flow series of the D-day has been determined, the low flow discharge of the D-day can be estimated according to any given return period. It should be noted that the research is station dependent on this analysis. The low flow-duration-frequency curves were therefore obtained at the base of gauging station. The low flow-duration-frequency curves are powerful tools for many applications, but particularly for engineering practice. An engineer may get any discharge of the low flow-duration-frequency curves from any low flow model. The fraction of non-zero flows in this river basin is always 100 per cent allowing one to measure up to 100-year return cycle D-day low flow discharges.

**Referee #1 Comment:**

(b) how low flow is defined (based on the results I believe as the minimum annual flow but this is not clear from the methods section),

**Authors Response:**

Thank you for your recommendation. We have revised the manuscript in lines 139-143, page 5:

The low flow indicator applied to the available time series is the minimum low flow for weekly or 7-day low flow. For calendar years, the annual indicators were taken out. Low flow index chosen in our study is mean annual minimum flow on a 7-day average (MAM7) basis. For this study, two indicators are chosen, which characterise low flow differently: Q95 and MAM7. Both parameters are less sensitive to measurement errors than the minimum discharge.

The MAM7 represents the annual minimum of the mean on seven consecutive days of daily flows. It is used in the Netherlands, in Germany and also in the United-States and United Kingdom. The percentile 95 is the flow that is exceeded 95% of the time. This indicator is spread mainly in Europe for his pertinence in numerous fields of water resources management.

The average of the annual series of minimum 7-day average flows known as Mean Annual 7-day Minimum flow (MAM7) and is used in some countries, e.g. the UK for abstraction licensing. The 7-day period covered by MAM7 eliminates the day-to-day variations in the artificial component of the river flow. Also, an analysis based on a time series of 7-day average flows is less sensitive to measurement errors. At the same time, in the majority of cases, there is no significant difference between 1-day and 7-day low flows.

**Referee #1 Comment:**

(c) for which variable/events return periods were determined

**Authors Response:**

The frequency analysis consists in the adjustment of a statistical law to the hydrological observations for each station. The objective is to calculate the critical low flow ($Q_T$) that corresponds to a given return period ($T$). $T$ is defined as the mean time between two occurrences of low flows. To do so, we used probabilistic models. These models are mathematical formulations that aim at simulating natural hydrological phenomenon such as probabilistic processes based on the probabilistic analysis of the considered random variables (in this study, Q95 and MAM7).

**Referee #1 Comment:**

(d) whether the determination of return periods relies on empirical or theoretical distributions,

**Authors Response:**

The return period relies on theoretical distribution.

**Referee #1 Comment:**

(e) which drought characteristics were analysed in the below threshold drought analysis,

**Authors Response:**

To identify streamflow drought occurrences, we used a threshold level approach, a methodology introduced by Yevjevich (1967) and widely used in recent studies. The threshold levels (also referred to as truncation levels) were derived from the flow duration curve as the flow equalled or exceeded for 70% (Q70), 80% (Q80) and 90% (Q90) of the time, as indicators of moderate, severe and extreme streamflow droughts, respectively.

The thresholds were selected in order to balance the appearance of multi-year droughts and zero-drought years (when the flow never falls below the threshold level in a year), both essential features when choosing a consistent threshold level. A significant advantage of this methodology in comparison with the use of standardised drought indices is that it allows quantification of the deficit volume, which is a vital characteristic in water resources management. A drought event starts when the flow falls below the threshold and ends when the flow exceeded the threshold level. The threshold level approach is mostly used to estimate a hydrological drought. A sequence of drought events can be obtained using the streamflow and threshold levels. Each drought event is characterised by duration, deficit-volume and time-off.

**Referee #1 Comment:**
(f) whether a short pooling window of 7 days (l.220) actually guaranteed independence of events,

**Authors Response:**
The minimum seven days average discharge was obtained for each gauge for each dry year by first smoothing hydrographs with a 7-day moving average filter. Given that the use of the threshold level method applied on daily data introduces the problem of dependency between deficits and minor drought events, the streamflow time series were smoothed with a 7-day moving average prior to the threshold level calculation, following the World Meteorological Organization (WMO) (2008) recommendation.

**Referee #1 Comment:**
(g) whether minor droughts were removed or not (methods section says yes, results section says no (l. 297)),

**Authors Response:**
We have revised the manuscript in lines 278-280, page 9-10: In this paper, the 7-day moving average was applied as a pooling procedure to obtain smooth data. Through these methods, the mutually dependent drought events will combine into individual and independent drought events (Fleig et al., 2006). The minor drought events will be eliminated or combined with individual drought events automatically (Yahiaoui et al., 2009).

**Referee #1 Comment:**
(h) what the storage-draft rate method does and what kind of storage it refers to (an illustration of the concept would help).

**Authors Response:**
Thank you for your recommendation. We have revised the manuscript in lines 282-290, page 10: The water supply or inflow is depending on low flow characteristics in the stream. When the inflow rate is less than the outflow (demand) rate, the maximum amount of water drawn from storage is the cumulative difference between supply and demand volumes of dry seasons. In channel storage, the function of both outflow and inflow discharge can be considered under two categories as prism and wedge storage. The water surface flow in the channel is not only unparallel to channel bottom but also varies with time. The storage, which is the maximum cumulative deficiency in any dry season, is obtained from the maximum difference in the ordinate between the mass curve of water supply and demand. Thus, the storage required can be expressed as in Eq. (10):

$$S = Maximum\ of\ (\Sigma V_D - \Sigma V_S), \qquad (10)$$

Where, $V_D$ = Demand Volume; $V_S$ = Supply volume.

The minimum storage draft rate was determined by using the mass curve of low flow at a monthly interval (Bharali, 2015). Although specific evaluation of storage requirements is essential for design, reconnaissance planning can frequently be facilitated by using draft-storage curves based on low flow frequency analysis. Alrayess et al. (2017) determined the capacity of river storage by the mass curve method. The mass curve has many useful applications in the design of storage capacities, such as to determine the reservoir storage capacity and flood routing (Gao et al., 2017). The procedure for the mass curve method has the following steps; first, construct a mass curve of the historical streamflow (monthly streamflow); determine the slope of the cumulative draft line for the graphical scales; next, superimpose the cumulative draft line on the mass curve; lastly, measure the largest intercept between the cumulative draft line and the mass curve (Figure 1). The term draft rate refers to the residual flow to be maintained at downstream and the user demand. The storage means active storage that is available for inflow regulation.

[Figure]

Figure 1. Minimum storage required using mass curve analysis

The estimation of the storage draft rate in this study will determine the minimum storage of a river to sustain the water supply during low flows and droughts. The mass curve of the monthly low flow rate is used in this analysis to obtain the minimum storage rate of the river. The mass curve analysis of low flow for the duration of January to December plotted against duration for recurrence interval of 10-year. The cumulative draw off corresponds to a constant draft rate of 50% of the mean annual flow and connected by a straight line. The slope of the line represents the average rate of flow that can be maintained between time. Thus, the slope of the straight line joining the starting point and the last points of the mass curve represents the average of discharge- over the whole period of plotted records.

**Referee #1 Comment:**
Results - The results section contains several paragraphs actually belonging to the methods and introduction sections (e.g. l. 246-250: and by the way I thought the trend analysis was performed using the non-parametric Mann-Kendall test and not linear regression).

**Authors Response:**
Thank you for your recommendation. We have restructured the manuscript based on your comments.

**Referee #1 Comment:**
There is even a statement that belongs to the introduction describing the 'primary purpose' of this study (l. 260-261).

**Authors Response:**
We have revised the manuscript in lines 113-116, page 4.

**Referee #1 Comment:**
I would in some instances replace results presented in tables by figures. This particularly concerns l.300 -351. I would try to visualise these results instead of presenting them as plain text. E.g. number of events as barplots, durations, and deficits as boxplots for all stations. This would allow for a comparison across stations. In addition, you could also plot deficit time series per station to compare particular events.

**Authors Response:**
Thank you for your recommendation. We have restructured the manuscript based on your comments in Figure 4, 5 (page 27), Figure 6 and 7 (page 28).

**Referee #1 Comment:**
Discussion - The discussion presents a lot of material that in my opinion belongs to the introduction (l. 393-411). I would instead discuss the implications of your findings for water management in the region.

**Authors Response:**
Thank you for your recommendation. We have revised the discussion part based on your comments in lines 475-482, page 16.

**Referee #1 Comment:**
Conclusions - Instead of providing a summary of the methods, focus on the insights we gain from this study. Currently this seems to be: 'Based on the analysis of the study, the estimated minimum storage-draft rates for each station cannot meet the water demand during low flow at specific return periods, which is 10-year recurrence interval for this research.' (l. 448-449). Formulating conclusions will be easier once you have identified a clear research question.

**Authors Response:**
We have revised the conclusion according to your comments in lines 521-529, page 17.

**Referee #1 Comment:**
References - Should be carefully checked. There is at least one duplicate (Sarailidis et al. 2019), and I would consistently use lower caps for nouns (e.g. Bakanogullari et al. 2014).

**Authors Response:**
We have edited the references part.

**Referee #1 Comment:**

Language - The article needs editing with respect to the use of tense and sentence structure. Some redundant information can be removed (e.g. l. 102 and l. 107).

**Authors Response:**

We have removed the redundant information.

**Referee #1 Comment:**

Figures and Tables - Figure 1: I would indicate the locations of the dams mentioned in l.90-99 if they are important for the analysis. But I am still unsure whether the storage-rate refers to reservoir storage or something else. I would reduce the density of the stream network displayed in order to increase the distinctiveness of the colors.

**Authors Response:**

Response: Thank you for your recommendation. We have restructured the figure 1 based on your comments.

**Referee #1 Comment:**

Figure 2: Is this figure really needed?

**Authors Response:**

Thank you for your comment. Figure 2 shows the example of plotting position using Weibull distribution with the fitting distribution for station S01. This figure can increase the reader's understanding of why the fitted distribution should be conducted for extreme events.

**Referee #1 Comment:**

Figure 4: Increase legend font, provide one legend for all subplots not per subplot. What does the dark grey bar mean? Increase size of axis labels.

**Authors Response:**

Thank you for your recommendation. We have revised the manuscript in Figure 8, page 29.

**Referee #1 Comment:**

Table 1: Can in my opinion be removed as information is also contained in Figure 1.

**Authors Response:**

Thank you for your comment. Table 1 is consisting of detail information such as the size of the area, coordinate and river name.

**Referee #1 Comment:**

Table 2: Introduce in methods section, reference should be provided for each distribution.

**Authors Response:**

Thank you for your recommendation. We have revised the manuscript in Table 2, page 30.

**Referee #1 Comment:**

Table 4: It seems as if trends were not only determined over the whole period but also for very short time periods of 7 years. This sub period analysis does in my opinion not make sense. I think I would plot time series of mean, minimum, and maximum flow for each catchment to illustrate the trends and instead remove the sub period analysis.

**Authors Response:**

The trend analysis has been performed on every 8-years sub period to find any significant trend. When using the 10-year sub-period, the study area does not reflect any significant trend in streamflow and no relative different for 10-years sub-period.

**Referee #1 Comment:**

Table 10: Is this table related to Figure 4, and if so how or could it even be removed?

**Authors Response:**

Thank you for your recommendation. We have removed the Table 10.

**Referee #1 Comment:**

Minor points -

- Trend detection and attribution is a pretty active research area and I would not agree that we are 'beginning to pay more attention to trend analysis' (l. 118).
- A goodness-of-fit test rejects or non-rejects a hypothesis but does not 'accept a fit' (l. 163).
- The return period in a univariate setting is defined as $T=1/(1-p)$, where p is the non-exceedance probability, i.e. T it is not the probability of occurrence itself (l. 188).
- l.353-359: move material to introduction.
- l. 363-367: move to methods section
- No further editing suggestions are provided as the manuscript in my opinion needs to be completely revisited.

**Authors Response:**

Thank you for your comment. We have revised the manuscript.

The manuscript focuses on analysing streamflow in seven stations in the Selangor state (Malaysia). The paper is interesting and presents an acceptable analysis, however, in my opinion there are a few drawbacks in the paper, which can be eliminated by carrying out some major revisions following the list of comments below.
MAJOR PROBLEMS: My main concern refers to the trend analysis. First, why the trend analysis has been performed on 5 8-years sub-periods?

**Authors Response:**
The trend analysis has been performed on every 8-years sub-period to find any significant trend. When using the 10-years sub-period, the study area does not reflect any significant trend in streamflow and no relative different for 10-years sub-period.

**Referee #2 Comment:**
Then, are the trends statically significant? Recommendation: specify if the trends are statically significant and the confidence level considered. Finally, the authors applied both the MK and the Sen's Slope to evaluate the trend sign, but besides the trend sign it could be interesting to detect the trend magnitude. I suggest to apply the Sen's Slope for the evaluation of the slopes of the trends and the Mann–Kendall test for the assessment of the statistical significance.

**Authors Response:**
Thank you for your recommendation. We have revised the manuscript in Table 4: Trend analysis for time series period, page 31.

**Referee #2 Comment:**
Which is the influence of the dams on the results of this study?

**Authors Response:**
The dam is only for the application that would benefit from this result. The minimum storage required in this study is referred to the drainage basin stores that consists of the surface of significant quantities of water that may regulate the rate at which input feeds through to the output. The storage, which is the maximum cumulative deficiency in any dry season, is obtained from the maximum difference in the ordinate between the mass curve of water supply and demand. The estimation of the storage draft rate in this study will determine the minimum storage of a river to sustain the water supply during low flows and droughts. The mass curve of the monthly low flow rate is used in this analysis to obtain the minimum storage rate of the river. The minimum storage draft rate was determined by using the mass curve of low flow at a monthly interval (Bharali, 2015). Although specific evaluation of storage requirements is essential for design, reconnaissance planning can frequently be facilitated by using draft-storage curves based on low flow frequency analysis. Alrayess et al. (2017) determined the capacity of river storage by the mass curve method. The mass curve has many useful applications in the design of storage capacities, such as to determine the reservoir storage capacity and flood routing (Gao et al., 2017). The procedure for the mass curve method has the following steps; first, construct a mass curve of the historical streamflow (monthly streamflow); determine the slope of the cumulative draft line for the graphical scales; next, superimpose the cumulative draft line on the mass curve; lastly, measure the largest intercept

between the cumulative draft line and the mass curve (Figure 1). The term draft rate refers to the residual flow to be maintained at downstream and the user demand. The storage means active storage that is available for inflow regulation.

[Figure]

Figure 1. Minimum storage required using mass curve analysis

The estimation of the storage draft rate in this study will determine the minimum storage of a river to sustain the water supply during low flows and droughts. The mass curve of the monthly low flow rate is used in this analysis to obtain the minimum storage rate of the river. The mass curve analysis of low flow for the duration of January to December plotted against duration for recurrence interval of 10-year. The cumulative draw off corresponds to a constant draft rate of 50% of the mean annual flow and connected by a straight line. The slope of the line represents the average rate of flow that can be maintained between time. Thus, the slope of the straight line joining the starting point and the last points of the mass curve represents the average of discharge- over the whole period of plotted records.

**Referee #2 Comment:**
Can the authors better explain the aims of the paper?

**Authors Response:**
The primary purpose of this study are: (1) to arbitrate the trend analysis of streamflow for 40 years; (2) to determine the best-fitted distribution of probability for each station for low flow frequency analysis; (3) to evaluate the hydrological drought characteristics, including severity, duration and magnitude; (3) to determine the minimum storage draft rates in 7 catchments in Selangor region in Malaysia. This study is essential to understand the concept of low flow, drought characteristics, and the predictive significance of river storage-draft rates in managing sustainable water catchment. The results are useful for developing measures to maintain flow variability and can be used to develop policies for risk management.

**Referee #2 Comment:**

Finally, the authors simply describe the results present in the study, and not discuss those results in depth. The authors should try to improve the discussion to underline the added value of their work compared to other similar in the same area and in different areas of the world.

**Authors Response:**

Thank you for your comments. We have revised the manuscript in lines 445-493, page 15-16.

**Referee #2 Comment:**

MINOR COMMENT:

The English grammar, syntax and punctuation should be improved and I recommend professional proofreading by a native speaker.

**Authors Response:**

Thank you for your comments. We have submitted the last manuscript for the professional proofreading. We have attached the proofreading certificate in the attachment.

**Referee #2 Comment:**

Add some references for the Mann-Kendall test, for the Sen's slope estimator and for each distribution of Table 2.

**Authors Response:**

Thank you for your comments. We have revised the manuscript in Table 2, page 30.

**Referee #2 Comment:**

Some references in the text are missing in the references list: Kannan et al. (2018); Sarailidis et al. (2019b). In the references list the latter is a duplicate of Sarailidis et al. (2019a).

**Authors Response:**

Thank you for your comments. We have revised the manuscript.

**Referee #2 Comment:**

Figure 1: I think that this figure is not sufficiently informative and it must be greatly improved. Can the authors try to better identify the different sub-basins? Moreover, can the authors show the position of the dams?

**Authors Response:**

Thank you for your comments. We have revised the manuscript in Figure 1, page 25.

**Referee #2 Comment:**

Figure 3: please describe what the boxes and the whiskers mean. Which percentiles or interquartile ranges are represented?

**Authors Response:**

Thank you for your comments. We have revised the manuscript in lines 359-364, page 12. The existence of extreme values in the streamflow data may be determined using the Box plot method. The Box plot method is based on creating a so-called Box graph. This graph is the best possibility for geometrical visualisation of the distribution of random variables in some groups. These groups are created ordering the streamflow data between the extreme values, minimum and maximum. Each group has an equal number of data, and in mathematical statistic, this is called a quartile. The skewness describes the form of distribution of the random variables and measures both direction and degree of asymmetry of the distribution of the random variables.

**Attachment:**

[Figure]

PSUK Communications LTD
83 Ducie Street
Manchester, UK
M1 2JQ

Email: info@qualityproofreading.co.uk
Tel: (+44) 0330 822 0143
VAT: GB 227 3104 40
Company number: 09305305

**Certificate for Proofreading and Editing Services Undertaken**

16th April 2020

Authors:

Hasrul Hazman Hasan, Siti Fatin Mohd Razali, Nur Shazwani Muhammad, Zawawi Samba Mohamed, Fird

This document certifies that the manuscript titled "Assessment of probability distributions and minimum storage draft-rate analysis in the equatorial region" was proofread and edited by Proofreading Service UK.

The editor aimed to ensure that the author's intended meaning was not altered during the review.

All of the suggested amendments were tracked with the Microsoft Word "Track Changes" feature. Therefore, the author had the option to reject or accept each change individually.

Kind regards,

Proofreading Service UK

[revised manuscript text omitted]

---

## Referee Report (RR1)

**Review for manuscript "Assessment of probability distributions and minimum storage draft-rate in the equatorial region"**

**Authors:** Hasrul Hazman Hasan, Siti Fatin Mohd Razali, Nur Shazwani Muhammad, Zawawi Samba Mohamed, Firdaus Mohamad Hamzah

**Journal:** Natural Hazards and Earth System Sciences

**Summary**

The study by Hasan et al. focuses on low flows, drought, and minimum storage draft-rates in seven catchments in the Selangor region in Malaysia. The study consists of four types of analyses: (1) a non-parametric trend analysis on annual mean, minimum, and maximum flows using the Mann-Kendall and Sen's slope tests; (2) a low flow frequency analysis on annual minimum flow using the Lognormal 2P distribution; (3) an analysis of drought characteristics determined using a fixed drought threshold at the 90th flow percentile; and (4) the determination of minimum storage draft rates necessary to ensure sufficient water supply during low flow periods.

**General remarks**

The revised version of the manuscript in my opinion hardly addresses the major points risen by the two reviewers and highlighted by the editor and does not show significant improvement compared to the earlier version. I therefore have to re-iterate my previous criticism: (1) the study still does not seem to follow a clear aim and motivation and lacks the specification of a research question; (2) it still has an unclear structure and shows elements belonging to Introduction, Methods, Results, Discussion, Conclusions all over the place (i.e. not all introductory material is in the introduction,…); (3) the method descriptions are still confusing and it is hard to tell how the analysis was exactly done; (4) the trend analysis has been performed on sub periods instead of on the whole period which leads to the detection of spurious trends, which are probably rather attributable to internal variability/oscillations; (5) a novel aspect is missing, which leads to insignificant conclusions. I still do not think that this study is publishable in NHESS.

I again discuss some major points, which I feel have not been properly addressed in the revised version of the manuscript.

**Major points**

- **Abstract:** The abstract is missing a clear problem statement. The study region of interest should be mentioned. I would give it a clear structure by listing the four elements of the analysis: (1) trend analysis, (2) low flow frequency analysis, (3) drought analysis, and (4) storage draft rate analysis. The abstract should also include a short summary of the main findings and end with a concluding statement (this requires a clear problem statement at the beginning).
- **Introduction:** The introduction needs a clear research question and should introduce the problem and some background knowledge related to this research question (or questions). Currently, the introduction lists various statements related to low flows and droughts but does not tell a compelling story. The introduction would profit from a clear distinction between low flows, droughts, and water scarcity (for a discussion on these different concepts

see e.g. [*Van Loon et al.*, 2016]). In addition, a short introduction to the concept of 'storage rate' should be provided (e.g. does storage refer to reservoir storage or another type of storage?). I suggest to restructure the introduction as follows: (1) introduce why are droughts, low flows, and water scarcity important and what is the relationship between the three, (2) introduce factors influencing drought and water scarcity characteristics, (3) introduce the storage-draft rate concept and how this is related to drought, (4) provide a short introduction of study area and the problem you are trying to solve, (5) state research question, and (6) provide a short overview of methods used to answer this question.

- **Data:** The following specification is necessary: Are the streamflow time series natural or influenced by water abstraction and storage (at least some of them seem to be influenced)? It is still unclear whether reservoirs are present in the study region. None of them are indicated in Figure 1 as pointed out by both reviewers.
- **Methodology:** In my understanding, the analysis consists of four main steps: (1) Trend analysis of annual mean, maximum, and minimum flows, (2) low flow frequency analysis based on annual minimum flows, (3) analysis of drought characteristics for individual events, and (4) storage draft analysis. Is this correct. If this is what was actually done, I would restructure the methods section accordingly. It is unclear which types of variables are used for which type of analysis. I only figured out e.g. which variables were of interest in the trend analysis when I started to look at the tables presented in the Results section. The methods descriptions are confusing and unclear and include a lot of unnecessary detail instead of providing essential information. I do for example not understand why a detailed description of Flow Duration Curves is necessary (these were just used to determine the drought threshold, right?). In my opinion, the detailed description of the Mann-Kendall test can be removed and be replaced by an appropriate reference (l. 131-157). Instead, it should be specified (a) for which variable/events return periods were determined, (b) which drought characteristics were analyzed in the below threshold drought analysis, (c) I would add the informative illustration and description provided in the responses to the reviewers to illustrate the storage draft rate concept. Furthermore, the trend analysis should be performed on the whole period 1971-2017 instead of on sub periods of 8 years to avoid the detection of spurious trends.
- **Results:** The results section contains several paragraphs actually belonging to the methods and introduction sections (e.g. l. 323-327, 360-365 (in my opinion not necessary at all as it can be assumed readers know what a boxplot is)). There is even a statement that belongs to the introduction describing the 'primary purpose' of this study (l. 336-337). I would restructure according to the restructuring also suggested for the Methods section: (1) Results of trend analysis, (2) results of low flow frequency analysis, (3) results of drought characteristics analysis, and (4) results of storage rate analysis. And also here, it always needs to be clear which variables the results refer to.
- **Discussion:** The discussion presents a lot of material that in my opinion belongs to the introduction and the methods section (l. 459-484). I would instead discuss the implications of your findings for water management in the region.
- **Conclusions:** Instead of providing a summary of the methods, focus on the insights we gain from this study. Currently this seems to be: 'Based on the analysis of the study, the estimated minimum storage-draft rates for each station cannot meet the water demand during low

flow at specific return periods, which is 10-year recurrence interval for this research.' (l. 514). Formulating conclusions will be easier once you have identified a clear research question.

- **References:** Should again be carefully checked. I would consistently use lower caps for nouns (e.g. Bakanogullari et al. 2014).

- **Language:** I appreciate that the authors had their manuscript checked by an editing service. However, I think that the article needs another round of editing with respect to the use of tense and sentence structure.

- **Figures and Tables:**
  - **Most figures:** Increase legend font, provide one legend for all subplots not per subplot. Increase size of axis labels.
  - **Figure 1:** I would indicate the locations of the dams mentioned in l.90-99 if they are important for the analysis. But I am still unsure whether the storage-rate refers to reservoir storage or something else.
  - **Figure 3:** Indicate that outliers are not displayed?
  - **Table 6:** The p-values should lie in the range of [0,1]. Were the column names mixed up? I would indicate for which distributions and catchments, H0 of 'the distribution of the sample corresponds to the theoretical distribution' was rejected.
  - **Table 8:** can in my opinion be removed as you just focused on a threshold of Q90. By the way, I would talk about Q10, to consistently refer to non-exceedance probabilities throughout the paper.

**Minor points**

No further editing suggestions are provided as the manuscript in my opinion needs to be completely revisited.

**References used in this review**

Van Loon, A. F. et al. (2016), Drought in a human-modified world: Reframing drought definitions, understanding, and analysis approaches, *Hydrol. Earth Syst. Sci.*, *20*(9), 3631–3650, doi:10.5194/hess-20-3631-2016.

---

## Author Response (AR2)

The Editors
Natural Hazards and Earth System Sciences
EGU - European Geosciences Union e.V.
Philippe Courtial
Kastenbauerstr. 2
81677 Munich
Germany

Dear Editors,
**Re: Resubmission of manuscript "Assessment of probability distributions and minimum storage draft-rate analysis in the equatorial region", nhess-2020-105**

Thank you for the opportunity to revise our manuscript, nhess-2020-105. We appreciate the careful review and constructive suggestions from all reviewers. We believe that the manuscript is substantially improved after making the suggested comments and recommendations.

Following this letter are the reviewer comments with our responses in red colour, including how and where the text was modified on-page and line numbers. Changes made in the manuscript are marked using red colour. The revised manuscript was submitted to proofreading and editing services by IBP Editing Services. The revision has been developed in consultation with all co-authors, and each author has approved the final form of this revision.

Thank you for your consideration.

Sincerely,
Hasrul Hazman Hasan
First Author,
Department of Civil Engineering,
Faculty of Engineering & Built Environment,
Universiti Kebangsaan Malaysia, 43600 UKM Bangi, Selangor, Malaysia.
P99749@siswa.ukm.edu.my

Siti Fatin Mohd Razali
Corresponding author,
Department of Civil Engineering,
Faculty of Engineering & Built Environment,
Universiti Kebangsaan Malaysia, 43600 UKM Bangi, Selangor, Malaysia.
fatinrazali@ukm.edu.my

**Review for manuscript "Assessment of probability distributions and minimum storage draft-rate in the equatorial region."**
**Authors:** Hasrul Hazman Hasan, Siti Fatin Mohd Razali, Nur Shazwani Muhammad, Firdaus Mohamad Hamzah
**Journal:** Natural Hazards and Earth System Sciences

**Summary**

The study by Hasan et al. focuses on low flows, drought, and minimum storage draft-rates in seven catchments in the Selangor region in Malaysia. The study consists of four types of analyses: (1) a non-parametric trend analysis on annual mean, minimum, and maximum flows using the Mann- Kendall and Sen's slope tests; (2) a low flow frequency analysis on annual minimum flow using the Lognormal 2P distribution; (3) an analysis of drought characteristics determined using a fixed drought threshold at the $90^{th}$ flow percentile; and (4) the determination of minimum storage draft rates necessary to ensure sufficient water supply during low flow periods.

**General remarks**

The revised version of the manuscript in my opinion hardly addresses the major points risen by the two reviewers and highlighted by the editor and does not show significant improvement compared to the earlier version. I therefore have to re-iterate my previous criticism: (1) the study still does not seem to follow a clear aim and motivation and lacks the specification of a research question; (2) it still has an unclear structure and shows elements belonging to Introduction, Methods, Results, Discussion, Conclusions all over the place (i.e. not all introductory material is in the introduction,...); (3) the method descriptions are still confusing and it is hard to tell how the analysis was exactly done; (4) the trend analysis has been performed on sub periods instead of on the whole period which leads to the detection of spurious trends, which are probably rather attributable to internal variability/oscillations; (5) a novel aspect is missing, which leads to insignificant conclusions. I still do not think that this study is publishable in NHESS.

I again discuss some major points, which I feel have not been properly addressed in the revised version of the manuscript.

**Major points**

- **Abstract:** The abstract is missing a clear problem statement.

**Response:** Thank you for these observations. We have rewritten the abstract to better differentiate among the objectives and edited so that the methods are reflected in the results and the data support the conclusions.
We have revised the abstract based on your recommendation on page 1, lines 11-15.

The study region of interest should be mentioned.

**Response:** Thank you for your suggestion. We have revised the manuscript based on your recommendation on page 1, lines 15-17.

I would give it a clear structure by listing the four elements of the analysis: (1) trend analysis, (2) low flow frequency analysis, (3) drought analysis, and (4) storage draft rate analysis.

**Response:** Thank you for your suggestion. We agree with the reviewer. We have revised the manuscript based on your recommendation on page 1, lines 15-22.

The abstract should also include a short summary of the main findings and end with a concluding statement (this requires a clear problem statement at the beginning).
**Response:** Thank you for your suggestion. We agree with the reviewer. We have revised the manuscript based on your recommendation on page 1, lines 22-27.

- **Introduction:** The introduction needs a clear research question and should introduce the problem and some background knowledge related to this research question (or questions).

**Response:** Thank you for your suggestion. We have revised the manuscript based on your recommendation.

Currently, the introduction lists various statements related to low flows and droughts but does not tell a compelling story. The introduction would profit from a clear distinction between low flows, droughts, and water scarcity (for a discussion on these different concepts see e.g. [*Van Loon et al.*, 2016]).

**Response:** Thank you for your suggestion. We agree with the reviewer and have added the sentences in the Introduction section (Page 2 to 4, lines 42–100).

In addition, a short introduction to the concept of 'storage rate' should be provided (e.g. does storage refer to reservoir storage or another type of storage?).

**Response:** Thank you for your suggestion. We agree with the reviewer. We have revised the manuscript based on your recommendation on page 3 (line 80) to page 4 (line 100).

I suggest to restructure the introduction as follows: (1) introduce why are droughts, low flows, and water scarcity important and what is the relationship between the three,

**Response:** Thank you for your suggestion. We agree with the reviewer. We have revised the manuscript based on your recommendation on page 1 (line 29) to page 2 (line 60).

(2) introduce factors influencing drought and water scarcity characteristics,

**Response:** Thank you for your suggestion. We agree with the reviewer. We have revised the manuscript based on your recommendation on page 2 (line 62) to page 3 (line 90).

(3) introduce the storage-draft rate concept and how this is related to drought,

**Response:** Thank you for your suggestion. We agree with the reviewer. We have revised the manuscript based on your recommendation on page 3 (line 80) to page 4 (line 100).

(4) provide a short introduction of study area and the problem you are trying to solve,

**Response:** Thank you for your suggestion. We agree with the reviewer. We have revised the manuscript based on your recommendation on page 4, lines 99-113.

(5) state research question, and

**Response:** Thank you for your suggestion. We agree with the reviewer. We have revised the manuscript based on your recommendation on page 4, lines 118-122.

(6) provide a short overview of methods used to answer this question.

**Response:** Thank you for your suggestion. We agree with the reviewer. We have revised the manuscript based on your recommendation on page 4, lines 128-132.

• **Data:** The following specification is necessary: Are the streamflow time series natural or influenced by water abstraction and storage (at least some of them seem to be influenced)?

**Response:** Thank you for your suggestion. Three over seven stations is influenced by dam and the others are considered as natural streamflow time series without any influence by any dam.

It is still unclear whether reservoirs are present in the study region. None of them are indicated in Figure 1 as pointed out by both reviewers.

**Response:** Thank you for your suggestion. We agree with the reviewer. We have revised the manuscript based on your recommendation in Figure 1 and Table 1.

• **Methodology:** In my understanding, the analysis consists of four main steps: (1) Trend analysis of annual mean, maximum, and minimum flows, (2) low flow frequency analysis based on annual minimum flows, (3) analysis of drought characteristics for individual events, and (4) storage draft analysis. Is this correct. If this is what was actually done, I would restructure the methods section accordingly.

**Response:** Thank you for your suggestion. We agree with the reviewer. We have revised the manuscript based on your recommendation.

It is unclear which types of variables are used for which type of analysis. I only figured out e.g. which variables were of interest in the trend analysis when I started to look at the tables presented in the Results section. The methods descriptions are confusing and unclear and include a lot of unnecessary detail instead of providing essential information.

**Response:** Thank you for your suggestion. We agree with the reviewer. We revised the methods section based on your recommendation on page 5-6, lines 160-171 and Figure 2.

I do for example not understand why a detailed description of Flow Duration Curves is necessary (these were just used to determine the drought threshold, right?).

**Response:** We changed the sub-topic to 2.5 Threshold analysis. We have explained in detail about the threshold level that has been used in this study.

In my opinion, the detailed description of the Mann-Kendall test can be removed and be replaced by an appropriate reference (l. 131-157). Instead, it should be specified (a) for which variable/events return periods were determined,

**Response:** We have revised the manuscript based on your recommendation on page 6, lines 173-181.

(b) which drought characteristics were analysed in the below threshold drought analysis,

**Response:** We have revised the manuscript based on your recommendation on page 11, lines 326-327.

(c) I would add the informative illustration and description provided in the responses to the reviewers to illustrate the storage draft rate concept.

**Response:** Thank you for your suggestion. We agree with the reviewer. We have revised the manuscript based on your recommendation on page 10, lines 293-300 and Figure 3.

Furthermore, the trend analysis should be performed on the whole period 1971-2017 instead of on sub periods of 8 years to avoid the detection of spurious trends.

**Response:** Thank you for your suggestion. We have revised the manuscript based on your recommendation about the trend analysis by performed on the whole period of study, 1978 to 2017. Annual streamflow series trend analysis presents the overall view of the shift in systems of streamflow (Assefa and Moges, 2018). The Mann-Kendall test, Sen's slope, relative change within 40 years, maximum cumulative sum (CUSUM) with the year of change point and their value of $p$ using Pettitt test are displayed in Table 4.

- **Results:** The results section contains several paragraphs actually belonging to the methods and introduction sections (e.g. l. 323-327, 360-365 (in my opinion not necessary at all as it can be assumed readers know what a boxplot is)).

**Response:** We have revised the manuscript based on your recommendation by remove the unnecessary paragraphs in the results section and rewrite scientifically.

There is even a statement that belongs to the introduction describing the 'primary purpose' of this study (l. 336-337).

**Response:** Thank you for your suggestion. We agree with the reviewer. We have revised the manuscript based on your recommendation and removed the introduction statements in the result section.

I would restructure according to the restructuring also suggested for the Methods section: (1) Results of trend analysis, (2) results of low flow frequency analysis, (3) results of drought characteristics analysis, and (4) results of storage rate analysis. And also here, it always needs to be clear which variables the results refer to.

**Response:** Thank you for your suggestion. We agree with the reviewer. We have revised the manuscript based on your recommendation for restructuring the Method section.

- **Discussion:** The discussion presents a lot of material that in my opinion, belongs to the introduction and the methods section (l. 459-484). I would instead discuss the implications of your findings for water management in the region.

**Response:** Thank you for your suggestion. We agree with the reviewer. We have revised the manuscript based on your recommendation. We have combined the discussion section into the result section for a better understanding of the reader and clear presentation of all variable.

- **Conclusions:** Instead of providing a summary of the methods, focus on the insights we gain from this study. Currently, this seems to be: 'Based on the analysis of the study, the estimated minimum storage-draft rates for each station cannot meet the water demand during low flow at specific return periods, which is 10-year recurrence interval for this research.' (l. 514). Formulating conclusions will be easier once you have identified a clear research question.

**Response:** Thank you for your suggestion. We agree with the reviewer. We have revised the manuscript based on your recommendation and rewrite the conclusion section based on the result of this study.

- **References:** Should again be carefully checked. I would consistently use lower caps for nouns (e.g. Bakanogullari et al. 2014).

**Response:** Thank you for your suggestion. We have clearly checked and revised the references part.

- **Language:** I appreciate that the authors had their manuscript checked by an editing service. However, I think that the article needs another round of editing with respect to the use of tense and sentence structure.

**Response:** Thank you for your suggestion. The revised manuscript was submitted to proofreading and editing services by IBP Editing Services. The certificate was attached below.

- **Figures and Tables:**

- **Most figures:** Increase legend font, provide one legend for all subplots not per subplot. Increase size of axis labels.

**Response:** Thank you for your suggestion. We agree with the reviewer. We have revised the all figures based on your recommendation.

- **Figure 1:** I would indicate the locations of the dams mentioned in l.90-99 if they are important for the analysis. But I am still unsure whether the storage-rate refers to reservoir storage or something else.

**Response:** Thank you for your suggestion. We agree with the reviewer. We have revised Figure 1 based on your recommendation.

- **Figure 3:** Indicate that outliers are not displayed?

**Response:** Thank you for your suggestion. We agree with the reviewer. We have revised Figure 3 (current in Figure 5) based on your recommendation.

- **Table 6:** The p-values should lie in the range of [0,1]. Were the column names mixed up? I would indicate for which distributions and catchments, H0 of 'the distribution of the sample corresponds to the theoretical distribution' was rejected.

**Response:** Thank you for your suggestion. We agree with the reviewer. We have revised Table 6 based on your recommendation.

- **Table 8:** can in my opinion be removed as you just focused on a threshold of Q90. By the way, I would talk about Q10, to consistently refer to non-exceedance probabilities throughout the paper.

**Response:** We agree with the reviewer. We have removed Table 8 based on your recommendation.

**Minor points**

No further editing suggestions are provided as the manuscript in my opinion needs to be completely revisited.

**References used in this review**

Van Loon, A. F. et al. (2016), drought in a human-modified world: Reframing drought definitions, understanding, and analysis approaches, *Hydrol. Earth Syst. Sci.*, *20*(9), 3631–3650, DOI:10.5194/hess-20-3631-2016.

Attachment

[Figure]
 **IBP Editing Services**

**LANGUAGE EDITING**
**CERTIFICATE**

This document certifies that the manuscript listed below was edited for proper English language, grammar, punctuation, spelling, and overall style by a highly-qualified English-speaking editor at IBP Editing Services.

**Manuscript title:**
Assessment of probability distributions and minimum storage draft-rate analysis in the equatorial region

**Authors:**

Hasrul Hazman Hasan[1], Siti Fatin Mohd Razali[1], Nur Shazwani Muhammad[1], Firdaus Mohamad Hamzah[2]

**Date Issued:**
3rd October 2020

This document certifies that the manuscript listed above was edited for proper English language, grammar, punctuation, spelling, and overall style. Neither the research content nor the authors' intentions were altered in any way during the editing process. Documents receiving this certification should be English-ready for publication; however, the author has the ability to accept or reject our suggestions and changes. If you have any questions or concerns about this document or certification, please contact info@ibpeditingservices.com

IBP Editing Services (JM0753768-A)

info@ibpeditingservices.com

ibpeditingservices.com

**Review for manuscript "Assessment of probability distributions and minimum storage draft-rate in the equatorial region."**
**Authors:** Hasrul Hazman Hasan, Siti Fatin Mohd Razali, Nur Shazwani Muhammad, Firdaus Mohamad Hamzah
**Journal:** Natural Hazards and Earth System Sciences

**General remarks**
In the previous review, my main comments referred to the trend analysis and the figure quality. I recognize that in this revised manuscript almost all my suggestions have been addressed and incorporated. Therefore I think that, in the present form, the paper can be published in NHESS.

**Response:** We would like to thank you for your previous suggestion and appreciate your support in our paper.

**Review for manuscript "Assessment of probability distributions and minimum storage draft-rate in the equatorial region."**
**Authors:** Hasrul Hazman Hasan, Siti Fatin Mohd Razali, Nur Shazwani Muhammad, Firdaus Mohamad Hamzah
**Journal:** Natural Hazards and Earth System Sciences

**General remarks**
Dear Editor, Dear Authors,
I read with interest this manuscript for possible publication in NHESS journal.
I went through the comments of the previous referees and of the Editor and I can confirm that the authors made a significant and appreciable effort in responding to the requests. However, in reading the final manuscript I had the following observations:

Response: We would like to thank you for your constructive comments. We agree with most of the suggestions and, therefore, we modified the manuscript to take on board your comments.

• Manuscript is a bit too long. Description of the methodology can be made shorter. For example, description of the box plot method (L360-370) is not really necessary, as also the definition of the quartile (this is just an example).

Response: Thank you for your suggestion. We agree with the reviewer. We have revised the manuscript based on your recommendation by removed the description of boxplot method.

• Hydrological drought is normally defined based on the variation of precipitation regimes with respect to the expected volumes. In this study the hydrological drought is analyzed with respect to the streamflow variable and the connection with the low flow. Then, the changes in the observed streamflow can be due to both variations in precipitation and in the processes that determine the rainfall-runoff transformation, which are highly non-linear. You should emphasize this difference (precipitation vs streamflow); analyses of rainfall series would have been appropriate, however I suggest at least to include and discuss about the possible connection of the precipitation shortage with the negative trend of streamflow. This can be the main cause (L320).

Response: Thank you for your suggestion. We agree with the reviewer. We have revised the manuscript based on your recommendation in pages17 -18, lines 526-537.

• Are there any gaps in the streamflow time series? If so, how did you manage them?

Response: Thank you for your comments.
The streamflow time series are not containing any gap in 40-years historical records. The data were selected to cover the whole Selangor region with a common period, from 1978 to 2017. The criteria for selecting series were that the records should be, as far as possible, unaffected by human-induced changes in the basin and that the records should be continuous and as long as possible.

• L372-373: correct, but how can you tell this? It is not really supported by the results/discussion.

Response: Thank you for your comments. We have revised the manuscript based on your recommendation on page 14, lines 409-412.

• Figure 3: please, make the low flow axes in a more readable unit (es. m/h or mm/s)

Response: Thank you for your suggestion. We agree with the reviewer. We have revised Figure 3 (currently in Figure 5) based on your recommendation.

**Review for manuscript "Assessment of probability distributions and minimum storage draft-rate in the equatorial region"**
**Authors:** Hasrul Hazman Hasan, Siti Fatin Mohd Razali, Nur Shazwani Muhammad, Firdaus Mohamad Hamzah
**Journal:** Natural Hazards and Earth System Sciences

**General remarks**
This paper is aiming to understand the concept of low flow, to estimate hydrological drought characteristics, and the predictive significance of river storage-draft rates in operational water resources management in Selangor state, Malaysia. Hence, it uses four types of analyses: (1) a non-parametric trend analysis on annual mean, minimum, and maximum flows using the Kendall and Sen's slope test, (2) a low flow frequency analysis on annual minimum flow using the theoretical distributions, (3) an analysis of drought characteristics determined using a fixed drought threshold at the 90th flow percentile, and (4) the determination of minimum storage draft rates necessary to ensure sufficient of water supply during low flow periods. The paper is a new research study, but all the sections apart from the Introduction Section are written as a technical report. Hence, the application research part needs improvements and corrections to verify the novelties of the method employed in the study area. Based on this general comment the following points should be addressed and clarified.

**Response: We would like to thank you for your constructive comments. We agree with most of the suggestions and, therefore, we will modify the manuscript to take on board your comments.**

1. Uniformity of the Sections. I am having difficulties to connect all individual sections in a unified and complete framework. Several analyses are performed individually but the results of the sections are not used in the other sections. This makes the manuscript difficult to follow. For example what is the use of 2.4 Section in the subsequent sections? Again how FDCs (Section 2.5) are used in the manuscript? Please justify these issues on the revised manuscript.

**Response:** Thank you for your suggestion. We agree with the reviewer.

We have revised the manuscript based on your recommendation and connect all the sections for better understanding. The methodology framework for this study was constructed for all analysis. Sub-section 2.4.1 was combined to section 2.4 for estimated the return period of low flow after selection of the distribution that best fits the 7-day low flow data sample. Section 2.5 was changed to threshold level method that consists with developed the flow duration curve (FDC) for determining the 90$^{th}$ percentiles of streamflow series.

From the streamflow time series in section 2.4, when a set of streamflow ranked from highest to lowest is plotted against a log-transformed return period, or log-transformed exceedance frequency, a line is obtained. The slope of the obtained line is positive (for return period) and negative (for FDC). The fixed threshold is derived from the flow duration curve (FDC) based on the entire record period. The variable threshold approach is adapted to detect streamflow deviations for both high- and low-flow seasons. Lower than average flows during high-flow seasons may be important for later drought development. Streamflow deficits were calculated using the threshold level method, according to which a deficit is defined as a period when the flow is below a predefined discharge. The deficit duration is defined as the period when the flow is below the threshold. The volume of the deficit is defined as the sum of discharges for the corresponding deficit duration, as the intensity of deficit is defined the ratio

between the volume and the duration of the deficit. Finally, the last characteristic is the minimum flow of a deficit. In this study, discharge values resulting from Q90 quantiles from the flow duration curve (FDC) were used as thresholds.

2. Trend Analysis. Table 4 presents the results of the trend analysis for 8-year time interval and for the complete dataset (40 years). How these periods are selected and why? My advice to the authors is to use tests to identify significant step changes in the streamflow data (non-parametric tests (i.e. Distribution Free CUSUM) and/or parametric tests (i.e. Cumulative Deviation, Worsley Likelihood Ratio) and then to apply the trend tests in the identified time periods (if any) (Kundzewicz and Robson, 2004).

**Response:** Thank you for your suggestion. We have been revised the manuscript and redo the streamflow trend analysis for all stations and whole study periods (40 years). The average annual streamflow is analysed using the Mann-Kendall test, and significant trends and distribution changes are discussed. The trend slope is calculated using the Sen's slope estimator, which produces the amount of change in trends. Finally, the change points in the long-term streamflow data are identified using the CUSUM test, and the changes in streamflow before and after the change points are explored using Pettitt's test. These research methods are used to determine long-term streamflow trend changes in 7 stations and the trend changes in spatial variability.

| Station | Record Length | Mann-Kendall | Sen's Slope | Relative Change Within the Record (%) | Maximum Cumulative Sum (CUSUM) | Change Point (Year) | Value of $p$ (Pettitt's test) |
|---------|---------------|--------------|-------------|----------------------------------------|--------------------------------|---------------------|-------------------------------|
| S01 | 1978 - 2017 | 0.03 | 0.30 | 36.51 | 6 | 1996 | 0.1215 |
| S02 | 1978 - 2017 | 0.00 | 0.15 | 21.80 | 14 | 1997 | 0.0004 |
| S03 | 1978 - 2017 | -0.46 | -0.02 | -20.00 | 8 | 2006 | 0.1295 |
| S04 | 1978 - 2017 | 0.03 | 0.02 | 43.47 | 8 | 2007 | 0.0845 |
| S05 | 1978 - 2017 | 0.62 | 0.06 | 12.05 | 4 | 2005 | 0.4469 |
| S06 | 1978 - 2017 | -0.35 | -0.06 | -55.56 | 8 | 2009 | 0.0086 |
| S07 | 1978 - 2017 | 0.14 | 0.20 | 39.22 | 8 | 2005 | 0.2286 |

3. Section 2.4. Please justify the use of this section. Based on the distribution fitting I would guess to connect this section with section 2.6. Furthermore, a discussion is needed for the estimation method of distribution parameters. I would like to see in the revised manuscript a comparison (or a discussion) of methods for selecting the best method (i.e method of moments, L-moments, maximum likelihood, maximum goodness-of-fit estimation method). Please address these critical issues in the revised manuscript.

**Response:** Thank you for your suggestion. We agree with the reviewer. We have revised the manuscript based on your recommendation. The justification of Section 2.4 (currently section 3.2, page 8, line 223-239) and page 14, line 418-429.

The least-squares method uses mathematical formulas to determine the parameters of an empirical distribution, such as the slope and intercept of the distribution. A best fit is achieved when the sum of squares of all deviations between the observed point and some theoretical function is minimised. The function is calculated for each point, and then the difference between the observed and calculated is squared such that the sum is minimised. This method has gain popularity and is especially useful if the theoretical function can be made linear. For large sample sizes, method of maximum likelihood is superior to others since the resulting estimators of population parameters are considered to be more efficient and accurate.

4. Section 2.5. Please provide information on threshold selection. Why the authors select a fixed threshold (90th percentile). Why a variable threshold method is not selected for this study (e.g. Van Loon, 2015)? I would expect from the authors to use at least a monthly varying threshold for this type of presented analysis. Furthermore, please discuss the effect of pooling procedure and the selected threshold in the derived results. A sensitivity analysis using different pooling procedures and thresholds could exemplify the used methods.

**Response:** Thank you for your suggestion. We agree with the reviewer. We have revised the manuscript based on your recommendation in page 10 (line 302-308) and page 11 (line 327-335).

However, the threshold selection should be further analysed because it is not clear that Q90 should be used as a representative threshold for rivers in a monsoon climate. The time resolution, whether to apply a series of annual, monthly, or daily streamflow, depends on the hydrologic regime in the region of interest. The choice of threshold level influences both the number of events and the presence of multi-year droughts in the derived drought series. The within-year droughts neither a large amount of multi-year droughts nor a large number of years without any droughts should be included in the series as these can complicate an extreme value analysis. For short data series the use of very low threshold levels can be problematic, as the derivation of statistical properties of droughts require a certain minimum number of events. This study using 40-years streamflow record data for hydrological drought analysis. A drought starts when the streamflow falls below a threshold level, and the drought recovers when the streamflow returns above the threshold level. The duration (run-length, $d_i$), total deficit (run-sum, $v_i$) which is the sum of the deficits, and magnitude ($v_i/d_i$) of each drought event can be readily obtained.

A sensitivity analysis is out of scope from this study. This can be done for the further studies about the selection of hydrological drought indicator.

5. Section 2.6. The minimum storage draft rate was determined by using the mass curve of low flow at a monthly interval. Please explain the procedure in detail. I would guess that the draft rate could be estimated from section 2.4 for a 10-year return period using for example the sequent peak algorithm. Please address this issue on the revised manuscript.

**Response:** Thank you for your suggestion. We agree with the reviewer. We have revised the manuscript based on your recommendation in page 10, lines 287-300 and Figure 3.

6. Standardisation procedure of the used runoff indices (Q95, MAM-7d). In order to compare the results a standardisation procedure could be applied in the streamflow data.

**Response:** Thank you for your suggestion. We think this is out of the aims and purpose of this study.

The threshold level $Q_p$ as an index of hydrological drought, is chosen to represent the boundary between normal and usually low streamflow. This choice is based on the characteristics of the streamflow regime as a percentile from the flow duration curve and is frequently applied for both perennial and intermittent streams. For perennial streams, threshold levels are chosen between Q70% and Q95%, for intermittent streams. The choice of threshold might be in a number of ways and is amongst other a function of the type of water deficit. A compromise may have to be made between including events that can really be regarded as significant deficits and including enough events for analysing their characteristics. Kannan et al. (2018) indicated the flow duration curve could be divided into five zones, representing high flows (0-10%), humid conditions (10-40%), medium-range flows (40-60%), dry conditions (60-90%), and low flows (90-100%). To compare the results a standardisation procedure could be applied in the streamflow data is out of scopes from the aims of this study.

Minor Comments
7. A flow diagram presenting the steps of the analysis could be useful for international readers.

**Response:** Thank you for your suggestion. We agree with the reviewer. We have revised the manuscript based on your recommendation in page 5-6, lines 160-171 and Figure 2.

[revised manuscript text omitted]

---

## Author Response (AR3)

**Review for manuscript "Assessment of probability distributions and minimum storage draft-rate in the equatorial region."**
**Authors:** Hasrul Hazman Hasan, Siti Fatin Mohd Razali, Nur Shazwani Muhammad, Firdaus Mohamad Hamzah
**Journal:** Natural Hazards and Earth System Sciences

**Editor Decision:** Publish subject to minor revisions (review by editor) (25 Oct 2020) by Brunella Bonaccorso

**Comments to the Author:**
Dear Authors,
We have received the last comments by the two anonymous referees. As you can see, there are still some concerns related to the lack of:
- a clear focus on the specific objectives of the study, compared to the excessive details on the well-established statistical techniques (Referee #3);
- a proper justification of the selected threshold Q90 and a standardization procedure of the runoff indices (Referee #4).
I overall agree with the referees and strongly invite you to take their suggestions properly into account in your final revision. In addition, the introduction still needs some improvements. For instance, the authors still confuse drought, which is a natural phenomenon, with water scarcity, which is due to anthropogenic causes.

We are grateful to the editor for his/her time and suggestions in helping to improve the manuscript.

**Major points**

Therefore, I suggest to delete the following sentences:
**LL 35-38**: "The main factor involved in hydrological drought is climate change and anthropogenic activities of surface water resources. The hydrological drought assessment gives a good interpretation of the water surface of the hydrological cycle. Hydrological drought also allows the incorporation of spatial details that impact internal storage and soil, vegetation and terrain characteristics."
**LL 46-47**: "Drought is most frequently the consequence of climate change and human activities in the particular area or regions."
**LL 62-65**: "The hydrological drought design system is somewhat complicated and susceptible to catchment characteristics or climate, and a combination of the two variables (Loon et al., 2015; Mohammed and Scholz, 2018; Zhai and Tao, 2017). Precipitation and temperature are two main factors among different environmental factors that mainly determine the climate model and antecedent situation for hydrological drought events (Joetzjer et al., 2013)."
**Response:** Thank you for your suggestion. We have deleted those sentences.

Also, rephrase **LL 47-48** as follows: "Human activities and poor management of water resources can lead to water scarcity, which could be exacerbated by drought."
**Response:** We have reworded the writing in the revised manuscript, Line 44-45.

Finally, I have read the revised manuscript and found out that there are still some grammar and language issues. Therefore I recommend checking the manuscript carefully to eliminate errors and to always use the past tense when describing what you attempted and gained.

Some examples are:

**L105**: replace "where" with "when"

**L118**: increasing or decreasing?

**L234**: I suggest "the values of the variables" instead of "the return values"

**L306**: Delete "objective"

**L333**: I suggest "have occurred for less than 15 days" and "dependent" instead of "dependable"

**Response:** Thank you for your recommendation. The manuscript has been revised extensively and carefully checking any grammar and language errors or mistakes. Also, the revised manuscript was submitted again to proofreading services.

**Anonymous Referee #3**
**Review for manuscript "Assessment of probability distributions and minimum storage draft-rate in the equatorial region."**
**Authors:** Hasrul Hazman Hasan, Siti Fatin Mohd Razali, Nur Shazwani Muhammad, Firdaus Mohamad Hamzah
**Journal:** Natural Hazards and Earth System Sciences

**General Comments:**

My impression in that every time the authors make considerable effort in revising the manuscript to accomplish the review requirements, but still it lacks of a clear and concise focus, with many unnecessary details and the lack of links among concepts (mainly in the introduction). As I wrote in my last review, I suggest to 'simplify' and make the paper more focused on the specific targets, which are limited to the statistical analysis of streamflow data of the study area in terms of trend, low flow frequency and minimum storage draft-rate. The methodology does not offer novelties; the used methods are well established in statistical hydrology therefore they do not require a deep description. Just focus on what is new of your work, i.e. the results from the analysis on 40 years of data of the specific study area. It is clear that the interest is limited for the area under study. Additionally, some statements are generic and not supported by the results. Just avoid them or explain by making the correct connections. I write down here some examples of repetition, unnecessary details, not supported statements:

We are grateful to the reviewer for his/her time and suggestions in helping to improve the manuscript.

**Major points**

L14: substitute 'this study is essential (you still have to demonstrate it) with something link ' this study aims at …'
**Response:** Thank you for your suggestion. This sentence has been modified in Line 12 to 16, page 1.

L25: 'the results indicated the hydrological droughts have generally become more frequent …'; in terms of what? Never defined the hydrological drought so far.
**Response:** We have reworded the writing in Line 26-27.

L35-36: this concept is repeated many times in different part of the introduction (L35-36; L46-L48; L75-79): just discuss it once.
**Response:** The manuscript has been revised extensively by removing the repeated concept and have been discussed detail in Line 68-71.

L80-L89: Honestly, I don't see the need of this paragraph here; unnecessary details, just discuss very briefly the connection of water storage with the variables under your study. Instead, I would have expected a discussion on other studies that analyze historical streamflow to assess hydrological drought.
**Response:** In the revised manuscript, we have stated that in Line 98-101.

L149-150: 'Low flow refers to a stream's regime that indicate … annual cycle…. On the climate'. Delete or move.

**Response:** Thank you for your suggestion. This sentence has been deleted in the revised manuscript.

Remove eq. 10, 11, 12, 13. All these are well established applied statistical concepts.
**Response:** Thank you for your recommendation. In the revised manuscript, we have removed Eq. 10,11,12 and 13.

L308-L311: repetition of exactly same statements.
**Response:** This has been corrected in Line 280-281.

L355-356: "This variability …. At regional bases". Not supported by results; how can you tell this? You can just try to explain the results based on cause-effect concepts.
**Response:** In the revised manuscript, we have removed this sentence.

L386-388: Same here, It is not supported by results.
**Response:** We have reworded the writing and supported the justification using finding from the study by Abdullah and Nakagoshi (2006) in Line 356-357.

**Response:** In the revised manuscript, we have stated that the selection of Q90 as threshold level and the used of two methods in pooling procedure which is moving average and inter-event time.

The threshold level is to represent the boundary between normal and unusual low streamflow, which selected based on the characteristics of the streamflow regime. In this case, low flow indices, such as percentiles from the flow duration curve (FDC), are frequently being applied for perennial streams. The threshold level has been chosen to compromise between these two features. For short data series, the use of very low threshold levels can be problematic, as the derivation of statistical properties of droughts requires a certain minimum number of events. These considerations do not reveal a single preferable threshold level, and its selection, and hence the definition of drought, remains as a subjective decision. For perennial streams, Q90 as threshold levels from the FDC have regularly been employed (Pyrce, 2004; Wilby et al., 1994).

For example, the study by Pyrce (2004), indicated that the percentiles of Q90 and Q95 have commonly been employed as low-flow metrics. They represent warning levels and limiting conditions (Q90), as well as the bases for biological and ecological indices (Q95), and the limits for surface-water extractions and effluent discharges. Empirical Q90 and Q95 values were extracted from the available time series for each gauge. Wilby et al. (1994) have indicated the effects of these scenarios on several indices from an FDC with particular emphasis on low flows, which have been illustrated using the example of catchments in the United Kingdom. It appeared that the most affected flow indices are Q10 and Q50, while extreme low flows (Q90) are affected mostly by land use. Other researchers such as Edossa et al. (2010); Hisdal et al. (2001); Sung and Chung (2014) defined and applied Q90 as the low flow indices for perennial streams.

Therefore, three percentiles Q85, Q90 and Q95 are considered as an early assumption of probable thresholds level at early stage of study. It was found that there are too few low flow events and too short dry spells when Q95 is chosen as a threshold, while some quick flows are included in the low flow event sample for a threshold equal Q85. Therefore, the Q90 threshold was selected, acted as a trade-off between sample size for statistical inferences and the sample quality. The monthly varying threshold was defined in Demuth et al. (2000), could not be applied in this paper, because the high variability of monthly flow across years can cause the low flow in a month to be compensated by the high flow. This compensation will cause potential deficits on the annual scale were cancelled out. More details about the Threshold Level Method and the selection of the threshold can be found in Fleig et al. (2006) and Gustard and Demuth (2008).

When applying the threshold level method in time series of discharge with a daily time step, dependent and minor drought events are introduced. Dependent drought events occurred when two prolonged drought periods are interrupted by a short excess period, and minor drought events are events with sort duration and low severity. A great number of dependent and minor droughts affect the statistical frequency analysis of the drought characteristics. Therefore, in order to deal with this problem, two pooling procedures were used, namely, moving average (MA) and inter-event time (IT). It is clear that for all of the pooling procedures employed in this study, there are parameters that need to be defined. The parameters are the number of days of the filter in the MA method and the time criterion in the IT method (Fleig et al., 2006). In our study, the MA method, moving filter of a 7-day averaging interval is applied to the complete time series. In this way, the dependent and minor droughts are filtered out, producing a new smoothed time series. In the IT method, two drought events are pooled if they occurred less than a predefined number of days. The minor drought events less than 15 days were excluded, as this approach has been justified by the study of Sakke et al. (2016). More details about the three pooling procedure can be found in Gustard and Demuth (2008).

2) Standardization procedure of the used runoff indices. A simple standardization procedure based on the area could be applied in the final streamflow deficits for comparison between the study watersheds.

**Response:** Thank you for your suggestion. However, the standardization procedure of the used the runoff indices, is basically for regionalization analysis of hydrological drought. This method is not covered in this study as we are focusing on historical hydrological drought in individual watersheds, and identifying any correlation between hydrological drought and minimum storage in the specific river at specific catchment. The relationship between the low flow event and the watershed area was represented in Figure 5. This analysis demonstrated the physical characteristics of each catchment and the significant effects of low flow events.

We are grateful to the reviewer for his/her time and suggestions in helping to improve the manuscript.

[revised manuscript text omitted]